# DeepGRAND: Deep Graph Neural Diffusion

## Abstract

We propose the Deep Graph Neural Diffusion (DeepGRAND), a class of continuous-depth graph neural networks based on the diffusion process on graphs. DeepGRAND leverages a data-dependent scaling term and a perturbation to the graph diffusivity to make the real part of all eigenvalues of the diffusivity matrix become negative, which ensures two favorable theoretical properties: (i) the node representation does not exponentially converge to a constant vector as the model depth increases, thus alleviating the over-smoothing issue; (ii) the stability of the model is guaranteed by controlling the norm of the node representation. Compared to the baseline GRAND, DeepGRAND mitigates the accuracy drop-off with increasing depth and improves the overall accuracy of the model. We empirically corroborate the advantage of DeepGRAND over many existing graph neural networks on various graph deep learning benchmark tasks.

## 1 Introduction

Graph neural networks (GNNs) and machine learning on graphs (Bronstein et al., 2017; Scarselli et al., 2008) have been successfully applied in a wide range of applications including physical modeling (Duvenaud et al., 2015; Gilmer et al., 2017; Battaglia et al., 2016), recommender systems (Monti et al., 2017; Ying et al., 2018), and social networks (Zhang & Chen, 2018; Qiu et al., 2018). Recently, more advanced GNNs have been developed to further improve the performance of the models and extend their application beyond machine learning, which include the graph convolutional networks (GCNs) (Kipf & Welling, 2017), ChebyNet (Defferrard et al., 2016), GraphSAGE (Hamilton et al., 2017), neural graph finger prints (Duvenaud et al., 2015), message passing neural network (Gilmer et al., 2017), graph attention networks (GATs) (Veličković et al., 2018), and hyperbolic GNNs (Liu et al., 2019). A well-known problem of GNNs is that the performance of the model decreases significantly with increasing depth. This phenomena is a common plight of most GNN architectures and referred to as the oversmoothing issue of GNNs (Li et al., 2018; Oono & Suzuki, 2020; Chen et al., 2020).

### 1.1 Main contributions and outline

In this paper, we propose Deep Graph Neural Diffusion (DeepGRAND), a class of continuous-depth graph neural networks based on the diffusion process on graphs that improves on various aspects of the baseline Graph Neural Diffusion (GRAND) (Chamberlain et al., 2021b). At its core, DeepGRAND introduces a data-dependent scaling term and a perturbation to the diffusion dynamic. With this design, DeepGRAND attains the following advantages:

1. DeepGRAND inherits the diffusive characteristic of GRAND while significantly mitigates the over-smoothing issue.

2. DeepGRAND achieves remarkably greater performance than existing GNNs and other GRAND variants when fewer nodes are labelled as training data, meriting its use in low-labelling rates situations.

3. Feature representation under the dynamic of DeepGRAND is guaranteed to remain bounded, ensuring numerical stability.

**Organization.** In Section 2, we give a concise description of the over-smoothing issue of general graph neural networks and the GRAND architecture. A rigorous treatment and further discussions

showing the inherent over-smoothing issue in GRAND is given in Section 3. The formulation for DeepGRAND is given in Section 4, where we also provide theoretical guarantees on the stability and ability to mitigate over-smoothing of DeepGRAND. Finally, we demonstrate the practical advantages of DeepGRAND in Section 5, showing reduced accuracy drop-off at higher depth and improved overall accuracy when compared to variants of GRAND and other popular GNNs.

## 1.2 RELATED WORK

**Neural ODEs.** Chen et al. (2018) introduced Neural ODEs, a class of continuous-depth neural networks with inherent residual connections. Follow-up works extended this framework through techniques such as augmentation (Dupont et al., 2019), regularization (Finlay et al., 2020), momentum (Xia et al., 2021). Continuous depth GNN was first proposed by Xhonneux et al. (2020).

**Over-smoothing.** The over-smoothing issue of GNNs was first recognized by Li et al. (2018). Many recent works have attempted to address this problem (Oono & Suzuki, 2020; Cai & Wang, 2020; Zhao & Akoglu, 2020). A connection between the over-smoothing issue and the homophily property of graphs was also recognized by Yan et al. (2021).

**PDE-inspired graph neural networks.** Partial differential equations (PDEs) are ubiquitous in modern mathematics, arising naturally in both pure and applied mathematical research. A number of recent works in graph machine learning has taken inspiration from classical PDEs, including: diffusion equation (Chamberlain et al., 2021b; Thorpe et al., 2022), wave equation (Eliasof et al., 2021), Beltrami flow (Chamberlain et al., 2021a), etc. Such approaches have led to interpretable architectures, and provide new solutions to traditional problems in GNNs such as over-smoothing (Rusch et al., 2022) and over-squashing (Topping et al., 2022).

## 2 BACKGROUND

**Notation.** Let $\mathcal{G} = (\mathcal{V}, \mathcal{E})$ denote a graph, where $\mathcal{V}$ is the vertex set with $|\mathcal{V}| = n$ and $\mathcal{E}$ is the edge set with $|\mathcal{E}| = e$. For a vertex $u$ in $\mathcal{V}$, denote $\mathcal{N}_u$ the set of vertices neighbor to $u \in \mathcal{G}$. We denote $d$ as the dimensionality of the features, i.e, number of features for each node. Following the convention of Goodfellow et al. (2016), we denote scalars by lower- or upper- case letters and vectors and matrices by lower- and upper-case boldface letters, respectively.

## 2.1 GRAPH NEURAL NETWORKS AND THE OVER-SMOOTHING ISSUE

As noted by Bronstein et al. (2021), the vast majority of the literature on graph neural networks can be derived from just three basic flavours: convolutional, attentional and message-passing. The cornerstone architectures for these flavours are GCN (Kipf & Welling, 2017), GAT (Veličković et al., 2018) and MPNN (Gilmer et al., 2017), the last of which forms an overarching design over the other two. An update rule for all message passing neural networks can be written in the form

$$\boldsymbol{H}_u = \xi \left( \boldsymbol{X}_u, \bigoplus_{v \in \mathcal{N}_u} \mu(\boldsymbol{X}_u, \boldsymbol{X}_v) \right),$$

where $\mu$ is a learnable message function, $\bigoplus$ is a permutation-invariant aggregate function, and $\xi$ is the update function. In the case of GCN or GAT, this can be further simplified to

$$\boldsymbol{H}_u = \sigma \left( \sum_{v \in \mathcal{N}_u \cup \{u\}} a_{uv} \mu(\boldsymbol{X}_v) \right), \tag{1}$$

where $a$ is either given by the normalized augmented adjacency matrix (GCN) or the attention mechanism (GAT), $\mu$ is a linear transformation, and $\sigma$ is an activation function.

The learning mechanism behind GCN was first analyzed by Li et al. (2018), who showed that graph convolution is essentially a type of Laplacian smoothing. It was also noted that as deeper layers are stacked, the architecture risks suffering from over-smoothing. This issue makes features indistinguishable and hurts the classification accuracy, which goes against the common understanding that

the deeper the model, the better its learning capacity will be. Furthermore, it prevents the stacking of many layers and impair the ability to model long range dependencies.

Over the past few years, over-smoothing has been observed in many traditional GNNs, prompting a flurry of research into understanding (Oono & Suzuki, 2020; Cai & Wang, 2020; Yan et al., 2021) and alleviating (Luan et al., 2019; Zhao & Akoglu, 2020; Rusch et al., 2022) the issue.

## 2.2 Graph Neural Diffusion

GRAND is a continuous-depth architecture for deep learning on graph proposed by Chamberlain et al. (2021b). It drew inspiration from the heat diffusion process in mathematical physics, and follows the same vein as other PDE-inspired neural networks (Rusch et al., 2022; Eliasof et al., 2021; Chamberlain et al., 2021a). Central to the formulation of GRAND is the diffusion equation on graph

$$\frac{\partial \boldsymbol{X}(t)}{\partial t} = \text{div}[\boldsymbol{G}(\boldsymbol{X}(t), t)\nabla \boldsymbol{X}(t)], \tag{2}$$

where $\boldsymbol{G} = \text{diag}\,(a(\boldsymbol{X}_i(t), \boldsymbol{X}_j(t), t))$ is an $e \times e$ diagonal matrix giving the diffusivity between connected vertices, which describes the thermal conductance property of the graph. The architecture utilises the encoder-decoder design given by $\boldsymbol{Y} = \psi(\boldsymbol{X}(T))$, where $\boldsymbol{X}(T) \in \mathbb{R}^{n \times d}$ is computed as

$$\boldsymbol{X}(T) = \boldsymbol{X}(0) + \int_0^T \frac{\partial \boldsymbol{X}}{\partial t}(t)dt, \;\; \text{with } \boldsymbol{X}(0) = \phi(\boldsymbol{X}). \tag{3}$$

In the simplest case when $\boldsymbol{G}$ is only dependent on the initial node features, the differential in equation 3 simplifies to

$$\frac{\partial \boldsymbol{X}}{\partial t}(t) = (\boldsymbol{A}(\boldsymbol{X}) - \boldsymbol{I})\boldsymbol{X}(t), \tag{4}$$

where $\boldsymbol{A}(\boldsymbol{X}) = [(a(\boldsymbol{X}_i(t), \boldsymbol{X}_j(t), t))]$ is an $n \times n$ matrix with the same structure as the adjacency matrix of the graph. From now on, we omit the term $\boldsymbol{X}$ when writing $\boldsymbol{A}(\boldsymbol{X})$. The entries of $\boldsymbol{A}$ are exactly those in $\boldsymbol{G}$, and thus determines the diffusivity. Furthermore, $\boldsymbol{A}$ can be informally thought of as the attention weight between vertices. Building upon this heuristic, GRAND models the attention matrix $\boldsymbol{A}$ in equation 4 by the multi-head self-attention mechanism, where

$$\boldsymbol{A} = \frac{1}{h}\sum_{l=1}^{h} \boldsymbol{A}^l(\boldsymbol{X}) \tag{5}$$

with $h$ being the number of heads and the attention matrix $\boldsymbol{A}^l(\boldsymbol{X}) = (a^l(\boldsymbol{X}_i, \boldsymbol{X}_j))$, for $l = 1, \ldots, h$. With this specific implementation, which is called GRAND-l, we obtain $\boldsymbol{A}$ as a right-stochastic matrix with positive entries.

As has been pointed out by Chamberlain et al. (2021b); Thorpe et al. (2022), many GNN architectures, including GAT and GCN, can be formalised as a discretisation scheme of equation 2 if no non-linearity is used between the layers. The intuition behind the connection can readily be seen from equation 1, where each subsequent node data is computed similar or equal to a weighted average of the neighboring node features. This gives rise to the diffusive nature of these architectures, resonating with the work of Gasteiger et al. (2019), which asserts that diffusion improves graph learning.

The choice of using a time-independent attention function amounts to all the layers sharing the same parameters, making the model lightweight and less prone to over-fitting. The original authors claimed that this implementation solves the over-smoothing problem and performs well with many layers. However, our analysis shows that this is not the case.

## 3 Does GRAND suffer from over-smoothing?

Various works (Oono & Suzuki, 2020; Cai & Wang, 2020; Rusch et al., 2022) have attributed the occurrence of the over-smoothing issue to the exponential convergence of node representations. We formally define this phenomenon for continuous depth graph neural networks.

**Definition 1.** *Let $\boldsymbol{X}(t) \in \mathbb{R}^{n \times d}$ denote the feature representation at time $t \geq 0$. $\boldsymbol{X}$ is said to experience over-smoothing if there exists a vector $\boldsymbol{v} \in \mathbb{R}^d$ and constants $C_1, C_2 > 0$ such that for $\boldsymbol{V} = (\boldsymbol{v}, \boldsymbol{v}, \ldots, \boldsymbol{v})^\top$*

$$\|\boldsymbol{X}(t) - \boldsymbol{V}\|_\infty \leq C_1 e^{-C_2 t}. \tag{6}$$

This definition is similar in spirit to the one given by Rusch et al. (2022). We will show that the GRAND architecture suffers from over-smoothing.

**Proposition 1.** *Given a positive right-stochastic matrix $\boldsymbol{A} \in \mathbb{R}^{n \times n}$, there exists an invertible matrix $\boldsymbol{P} \in \mathbb{C}^{n \times n}$ such that*

$$\boldsymbol{P}^{-1}(\boldsymbol{A} - \boldsymbol{I})\boldsymbol{P} = \begin{pmatrix} 0 & 0 & \ldots & 0 \\ 0 & \boldsymbol{J}_1 & \ldots & 0 \\ \vdots & \vdots & \ddots & \vdots \\ 0 & 0 & \ldots & \boldsymbol{J}_m \end{pmatrix} \tag{7}$$

*where each $\boldsymbol{J}_i$ is a Jordan block associated with some eigenvalue $\beta_i$ of $\boldsymbol{A} - \boldsymbol{I}$ with $\operatorname{Re} \beta_i < 0$, and $\boldsymbol{P}$ can be chosen such that its first column is the vector $(1, 1, \ldots, 1)^\top$.*

The Jordan decomposition of $\boldsymbol{A}$ allows us to study the long term behaviour of GRAND through its matrix spectrum. The fact that the real part of all eigenvalues of $\boldsymbol{A} - \boldsymbol{I}$ is equal or lesser than $0$ will be essential in our analysis.

**Proposition 2.** *With the dynamic given by equation 3, equation 4, and equation 5, $\boldsymbol{X}$ experiences over-smoothing.*

**Remark 1.** *A random walk viewpoint of GRAND was given by Thorpe et al. (2022). Using this, the authors were able to show node representations in the forward Euler discretization of the GRAND dynamic given by*

$$\boldsymbol{X}(k\delta_t) = \boldsymbol{X}((k-1)\delta_t) + \delta_t(\boldsymbol{A} - \boldsymbol{I})\boldsymbol{X}((k-1)\delta_t), \text{ for } k = 0, 1, 2, \ldots, K \text{ and } \delta_t < 1,$$

*converges in distribution to a stationary state. Our spectral method can be adapted to this discretized setting (see Appendix D). We note that our analysis is strictly stronger through both the use of the $\infty$-norm and the exact exponential rate of convergence.*

Proposition 2 conclusively shows that GRAND-l still suffers from over-smoothing. Of course, if we consider more general variants of GRAND, the above arguments no longer hold. Although we can not rigorously assert that over-smoothing affect all implementations of GRAND, it can be argued that the occurrence of this phenomenon in GRAND intuitively makes sense since diffusion have the tendency to 'even out' distribution over time (further discussion can be found in Appendix B). Hence, a purely diffusion based model like GRAND is inherently ill-suited for deep networks.

## 4 DEEPGRAND: DEEPER GRAPH NEURAL DIFFUSION

### 4.1 MODEL FORMULATION

We propose DeepGRAND, a new class of continuous-depth graph neural networks based on GRAND that retains the advantage of diffusion and is capable of learning at much higher depth. It leverages a perturbation to the graph diffusivity $\boldsymbol{A} - (1 + \epsilon)\boldsymbol{I}$ and a scaling factor $\langle \boldsymbol{X}(t) \rangle^\alpha$ to make the model both more stable and more resilient to the over-smoothing issue.

Denote the $i$-th column of $\boldsymbol{X}$ by $\boldsymbol{X}_i$, and the usual 2-norm by $\|\cdot\|$. For constant $\alpha > 0$, we define the column-wise norm matrix $\langle \boldsymbol{X}(t) \rangle^\alpha \in \mathbb{R}^{n \times d}$

$$\langle \boldsymbol{X}(t) \rangle^\alpha = \begin{pmatrix} \|\boldsymbol{X}_1\|^\alpha & \|\boldsymbol{X}_2\|^\alpha & \ldots & \|\boldsymbol{X}_d\|^\alpha \\ \|\boldsymbol{X}_1\|^\alpha & \|\boldsymbol{X}_2\|^\alpha & \ldots & \|\boldsymbol{X}_d\|^\alpha \\ \vdots & \vdots & \ddots & \vdots \\ \|\boldsymbol{X}_1\|^\alpha & \|\boldsymbol{X}_2\|^\alpha & \ldots & \|\boldsymbol{X}_d\|^\alpha \end{pmatrix}, \tag{8}$$

The dynamic of DeepGRAND is given by equation 3, with $\frac{\partial \boldsymbol{X}}{\partial t}$ given by

$$\frac{\partial \boldsymbol{X}}{\partial t}(t) = \big(\boldsymbol{A} - (1 + \epsilon)\boldsymbol{I}\big)\boldsymbol{X}(t) \odot \langle \boldsymbol{X}(t) \rangle^\alpha, \tag{9}$$

where $\odot$ is the Hadamard product. Our proposed model thus represents an alteration to the central GRAND dynamics given by equation 4. With such modifications, DeepGRAND no longer has the form of a pure diffusion process like GRAND, while still retaining its diffusive characteristics: at every infinitesimal instance, the information from graph nodes are aggregated for use in updating feature representation.

## 4.2 THEORETICAL ADVANTAGES OF DEEPGRAND

To explain the theoretical motivation behind DeepGRAND, we first note that the convergence property of all nodes to a pre-determined feature vector is not necessarily an undesirable trait. It guarantees the model will remain bounded and not 'explode'. Our perturbation by $\epsilon$ serves to slightly strengthen this behaviour. Its purpose is to reduce the real part of all eigenvalues to negative values, making all node representations converge to $0$ as the integration limit $T$ tends to infinity.

We observe that it is in fact the rate of convergence that is the chief factor in determining the range of effective depth. If the model converges very slowly, it is clear we can train it at high depth without ever having to worry about over-smoothing. As such, if we can control the convergence rate, we would be able to alleviate the over-smoothing issue. The addition of the term $\langle \boldsymbol{X}(t) \rangle^\alpha \in \mathbb{R}^{n \times d}$ serves this purpose. As the node representations come close to their limits, this term acts as a scaling factor to slow down the convergent process.

An exact bound for the convergent rate of our model is presented in Proposition 3.

**Proposition 3.** *Assuming $\boldsymbol{A}$ is a right-stochastic and radial matrix. With the dynamic given in equation 9, we have the bound for each column $\boldsymbol{X}_i$*

$$\left((2+\epsilon)\alpha T + \|\boldsymbol{X}_i(0)\|^{-\alpha}\right)^{\frac{-1}{\alpha}} \leq \|\boldsymbol{X}_i(T)\| \leq \left(\epsilon\alpha T + \|\boldsymbol{X}_i(0)\|^{-\alpha}\right)^{\frac{-1}{\alpha}}. \tag{10}$$

The equation 10 shows that each column $\boldsymbol{X}_i$ is bounded on both sides by polynomial-like terms. Hence, it can no longer converge at exponential speed. The over-smoothing problem is thus alleviated. Note also that the right hand side of equation 10 converges to $0$ as $T$ goes to infinity. This ensures node representation will not explode in the long run and helps with numerical stability.

## 5 EMPIRICAL ANALYSIS

In this section, we conduct experiments to compare the performance of our proposed method Deep-GRAND and the baseline GRAND. We aim to point out that DeepGRAND is able to achieve better accuracy when trained with higher depth and limitted number of labelled nodes per class. We also compare our results with those of GRAND++ (Thorpe et al., 2022) and popular GNNs architectures such as GCN, GAT and GraphSage.

### 5.1 EXPERIMENTS

#### 5.1.1 DATA

**Cora** (McCallum et al., 2000) : A citation network consisting of more than 2700 publications with more than 5000 connections and 7 classes. Each node is a vector of 0s and 1s indicating the presence - absence of the corresponding words in a corpus of 1433 words.

**Citeseer** (Giles et al., 1998) : Similar to Cora, the Citeseer dataset is a network of 3312 scientific publications with 4732 connections and each publication is classified into 6 classes. The node vectors consist of 0s and 1s indicating the presence - absence of corresponding words in a library of 3703 words.

**Pubmed** (Sen et al., 2008) : Contains 19717 scientific publications related to diabetes from the Pubmed database separated into 3 classes. The network consists of 44338 connections and each node vector is the TF/IDF representation from a corpus of 500 unique words.

**CoauthorCS** (Monti et al., 2016) : Co-authorship graph of authors with publications in computer science fields. The nodes of the network represent the authors and two authors are connected if they share co-publish. Each node represents each author's papers keywords and the classes represents

the authors' most notable research fields or studies. This graph consists of 18333 nodes and 163788 edges with 15 different classes.

**Computers** (McAuley et al., 2015) : A co-purchase graph extracted from Amazon where each node represents a product and the edges represent the co-purchased relations. The node features are bag-of-words vectors extracted from the product reviews. The dataset consists of 13752 nodes with 491722 edges. All the nodes are classified into 10 different product categories.

**Photo** (McAuley et al., 2015) : Similar to Computers, the node features in the Photo dataset represents the bag-of-words vectors derived from the product review and nodes (products) are connected if frequently co-purchased. The dataset consists of 7487 nodes with 119043 edges. The nodes are classified into 8 classes representing 8 product categories.

### 5.1.2 SETUP

**Experiment setup**  For most of the experiments, we used the same set of hyperparameters as GRAND and only changed some settings for the ablation studies specified in sections 5.2, 5.3 and 5.4. Similar to GRAND-l, we denote the specific implimentation of DeepGRAND when $A$ is only dependent on the initial node features as DeepGRAND-l. The experiments are conducted on the common benchmarks for node classification tasks : Cora, Citeseer, Pubmed, Computers, Photo and CoauthorCS using our NVIDIA RTX 3090 graphics card.

One thing to note is that we have made slight adjustments to the original implementation of GRAND. Firstly, in the original work, the authors of GRAND included a learnable scaling term $\alpha$ in front of equation 4. Secondly, they also added the value of $X(0)$ scaled by a learnable scalar $\beta$ to the equation. The final GRAND dynamics used in the original implementation is indicated in equation 11.

$$\frac{\partial X}{\partial t}(t) = \alpha(A(X) - I)X(t) + \beta X(0) \tag{11}$$

The initial value $X(0)$ is treated as a source term added to the overall dynamics before each evaluation step of the ODE solver. This modification is task specific. For fair comparison of GRAND and DeepGRAND, we have removed this from the original implementation and all of our experiments.

### 5.1.3 EVALUATION

We aim to demonstrate the advantages of DeepGRAND by evaluating our proposed method using the following experiments:

**Ablation study with different depths (5.2)** : For this experiment, we trained variants of GRAND and DeepGRAND using the same integration limits $T$ across a wide range of values and compare the respective test accuracies.

- **For Cora, Citeseer and Pubmed** : We used the default planetoid splitting method available for Cora, Citeseer and Pubmed. For each $T$ value, we trained GRAND-l, GRAND-nl and DeepGRAND-l with 10 random seeds (weights initializations) per method then calculated mean and standard deviation of the test accuracies across the random seeds.
- **For Computers, Photo and CoauthorCS** : We used random splitting for the 3 remaining benchmarks. For each $T$ value, we trained all methods with 10 random seeds and calculated mean and standard deviation of the test accuracies across the random seeds.

**Ablation study with different label rates (5.3)** : For all of the benchmarks, we used random splitting and trained GRAND-l, GRAND-nl and DeepGRAND-l with different number of labelled nodes per class. For each experiment, we ran with 10 random seeds per method and calculated mean and standard deviation of the test accuracies across the seeds.

**Ablation study - effect of $\alpha$ (5.4)** : For this section we aimed to study the effect of the exponential $\alpha$ on the ODE solver's ability to integrate. For each $\alpha$ value, we trained DeepGRAND-l with 10 random seeds and calculated the mean and standard deviation of the test accuracies across the seeds.

**Note** : For hyper-parameters tuning of DeepGRAND-l, we reused most of the settings tuned for GRAND and only tuned the hyper-parameters that are specific to DeepGRAND - the $\alpha$ and $\epsilon$ values specified in equation 9.

**Summary** : For both experiment sets presented in Table 1 and Table 2, we observed that our proposed dynamics achieved a substantially higher test accuracies than the baseline method and is less prone to performance decay effect when trained with deep architectures (large depth $T$). Furthermore, the results in Table 3 reveals that DeepGRAND-l is also more robust when the number of labelled nodes per class is low.

## 5.2 DEEPGRAND IS MORE ADAPTABLE TO DEEP ARCHITECTURES

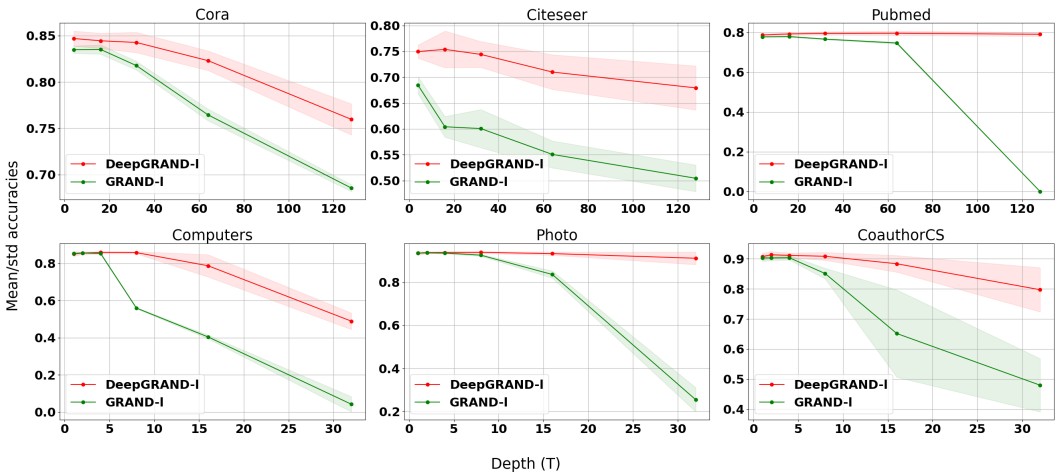

Figure 1: Effect of depth T on the performance of DeepGRAND-l and GRAND-l on Cora, Citeseer, Pubmed, Computers, Photo and CoauthorCS benchmarks. For the first three benchmarks, we used the default planetoid split to conduct our experiments. For the later three, we used random split.

Table 1: The means and standard deviations of test accuracies of DeepGRAND-l and variants of GRAND experimented with different depths on Cora, Citeseer and Pubmed under the default Planetoid split. NA: ODE solver failed. (Note: The results for GRAND++-l were imported from (Thorpe et al., 2022)).

| Dataset | Depth (T) | GRAND-l | GRAND-nl | GRAND++-l | DeepGRAND-l (ours) |
|---------|-----------|---------|----------|-----------|---------------------|
| Cora | 4 | $83.49 \pm 0.40$ | $83.31 \pm 0.27$ | $81.98 \pm 1.42$ | $\mathbf{84.69 \pm 0.23}$ |
| | 16 | $83.51 \pm 0.51$ | $82.53 \pm 0.35$ | $82.49 \pm 1.37$ | $\mathbf{84.45 \pm 0.29}$ |
| | 32 | $81.76 \pm 0.47$ | $81.66 \pm 0.25$ | $82.48 \pm 0.71$ | $\mathbf{84.26 \pm 0.44}$ |
| | 64 | $76.45 \pm 0.63$ | $78.70 \pm 1.08$ | $80.99 \pm 1.76$ | $\mathbf{82.13 \pm 0.67}$ |
| | 128 | $68.55 \pm 0.38$ | $74.60 \pm 0.51$ | $\mathbf{80.29 \pm 1.98}$ | $76.35 \pm 0.89$ |
| Citeseer | 4 | $68.45 \pm 1.66$ | $69.01 \pm 1.52$ | $72.58 \pm 3.79$ | $\mathbf{75.39 \pm 0.59}$ |
| | 16 | $60.42 \pm 2.00$ | $60.65 \pm 1.78$ | $73.84 \pm 2.66$ | $\mathbf{74.97 \pm 0.57}$ |
| | 32 | $60.06 \pm 3.69$ | $58.77 \pm 1.55$ | $73.29 \pm 1.29$ | $\mathbf{74.42 \pm 1.06}$ |
| | 64 | $55.06 \pm 2.61$ | $55.71 \pm 2.28$ | $\mathbf{72.81 \pm 2.18}$ | $71.00 \pm 1.08$ |
| | 128 | $50.45 \pm 2.56$ | $52.90 \pm 1.80$ | NA | $\mathbf{67.93 \pm 0.78}$ |
| Pubmed | 4 | $77.78 \pm 0.10$ | $77.50 \pm 0.34$ | $79.20 \pm 0.74$ | $\mathbf{79.61 \pm 0.98}$ |
| | 16 | $77.93 \pm 0.36$ | $77.76 \pm 0.15$ | $79.49 \pm 0.84$ | $\mathbf{79.52 \pm 0.56}$ |
| | 32 | $76.65 \pm 0.26$ | $78.41 \pm 0.37$ | $\mathbf{79.81 \pm 1.61}$ | $79.26 \pm 0.35$ |
| | 64 | $74.68 \pm 0.52$ | $73.31 \pm 1.23$ | NA | $\mathbf{79.08 \pm 0.59}$ |
| | 128 | NA | NA | NA | $\mathbf{78.74 \pm 0.46}$ |

We provide empirical evidence for our argument that the dynamics of DeepGRAND is more resilient to deeper architectures. Specifically, we compared the change in performance of both GRAND and DeepGRAND as the integration limit $T$ increases. In Table 1, for each $T$ value in $\{4, 16, 32, 64, 128\}$, we trained both methods for Cora, Citeseer and Pubmed datasets using the default planetoid splitting method with 10 random initializations. For the other benchmarks: Com-

Table 2: The means and standard deviations of test accuracies of DeepGRAND-l and variants of GRAND experimented with different depths on the Computers, Photo and CoauthorCS datasets under random split. NA: ODE solver failed. (Note: The results for GRAND++-l was imported from (Thorpe et al., 2022)).

| Dataset | Depth (T) | GRAND-l | GRAND-nl | GRAND++-l | DeepGRAND-l (ours) |
|---|---|---|---|---|---|
| Computers | 1 | $85.32 \pm 0.62$ | $85.67 \pm 0.14$ | $84.11 \pm 0.51$ | $\mathbf{85.87 \pm 0.30}$ |
| | 2 | $85.57 \pm 2.89$ | $85.02 \pm 0.67$ | $84.34 \pm 0.71$ | $\mathbf{85.78 \pm 0.28}$ |
| | 4 | $85.36 \pm 0.49$ | $84.23 \pm 0.52$ | $84.19 \pm 0.93$ | $\mathbf{85.54 \pm 0.49}$ |
| | 8 | $55.94 \pm 0.32$ | $52.14 \pm 0.44$ | $80.45 \pm 1.24$ | $\mathbf{85.12 \pm 0.32}$ |
| | 16 | $40.40 \pm 1.02$ | $42.11 \pm 3.03$ | $\mathbf{78.97 \pm 2.33}$ | $78.71 \pm 2.23$ |
| | 32 | $4.18 \pm 4.10$ | $8.19 \pm 5.13$ | $\mathbf{76.01 \pm 1.33}$ | $48.89 \pm 0.28$ |
| Photo | 1 | $93.55 \pm 0.20$ | $93.35 \pm 0.31$ | $92.93 \pm 0.84$ | $\mathbf{93.86 \pm 0.30}$ |
| | 2 | $93.65 \pm 0.24$ | $93.28 \pm 0.29$ | $\mathbf{93.89 \pm 0.22}$ | $93.80 \pm 0.28$ |
| | 4 | $93.52 \pm 0.32$ | $92.29 \pm 0.41$ | $93.54 \pm 0.38$ | $\mathbf{93.76 \pm 0.22}$ |
| | 8 | $92.57 \pm 0.47$ | $92.50 \pm 0.57$ | $93.01 \pm 0.52$ | $\mathbf{93.46 \pm 0.42}$ |
| | 16 | $83.58 \pm 1.67$ | $93.29 \pm 0.36$ | $92.69 \pm 0.61$ | $\mathbf{93.29 \pm 0.36}$ |
| | 32 | $25.45 \pm 5.63$ | $45.23 \pm 7.32$ | $\mathbf{92.94 \pm 0.90}$ | $91.13 \pm 0.95$ |
| CoauthorCS | 1 | $90.31 \pm 1.11$ | $91.12 \pm 0.44$ | $90.42 \pm 0.76$ | $\mathbf{91.36 \pm 0.16}$ |
| | 2 | $90.31 \pm 0.86$ | $90.79 \pm 0.14$ | $90.53 \pm 0.54$ | $\mathbf{91.17 \pm 0.65}$ |
| | 4 | $90.37 \pm 0.74$ | $89.90 \pm 0.75$ | $\mathbf{90.89 \pm 0.36}$ | $90.85 \pm 0.62$ |
| | 8 | $85.13 \pm 1.77$ | $87.18 \pm 1.04$ | $90.18 \pm 0.47$ | $\mathbf{90.74 \pm 0.65}$ |
| | 16 | $65.16 \pm 14.53$ | $77.68 \pm 3.74$ | $\mathbf{90.24 \pm 0.30}$ | $88.37 \pm 1.11$ |
| | 32 | $47.96 \pm 8.86$ | $49.71 \pm 12.08$ | NA | $\mathbf{79.73 \pm 2.35}$ |

puters, Photo and CoauthorCS, we splitted the datasets randomly and trained with 10 random initializations for each integration limit $T$ in $\{1, 2, 4, 8, 16, 32\}$ and recorded the results in Table 2. For each initialization, we used the default label rate of 20 labelled nodes per class as indicated in the original GRAND implementation. On all benchmarks, we observe that our proposed dynamics performed substantially better for larger values of $T$, indicating less performance degradation under the effect of over-smoothing.

### 5.3 DEEPGRAND IS MORE RESILIENT UNDER LIMITED LABELLED TRAINING DATA

In previous experiments, we showed that DeepGRAND outperforms the baseline GRAND when trained with high integration limit. In the following experiments, we aim to demonstrate that Deep-GRAND also achieves superior results with limited number of labelled nodes per class. For all of the benchmarks: Cora, Citeseer, Pubmed, Computers, Photo and CoauthorCS, we used grid search to find the optimal $T$ values and evaluate the performance under different numbers of labelled nodes per class. For each dataset, we experimented with 1, 2, 5, 10, 20 labelled nodes per class and compared the test accuracies between GRAND, GRAND++ and DeepGRAND-l. On top of GRAND variants, our results recorded in Table 3 also show that DeepGRAND outperforms some of the common GNN architectures like GCN, GAT and GraphSage. More experiment results where we used identical values of $T$ as those used in GRAND can be found in Appendix E. Even without using optimal $T$ values, DeepGRAND is still the top performing design more often than not.

### 5.4 EFFECT OF $\alpha$ ON THE DYNAMICS OF DEEPGRAND

The exponential $\alpha$ defined in our DeepGRAND dynamics directly influences the ability to integrate of the ODE solver. Specifically, a high $\alpha$ value (from $0.5$ to $1.5$) will likely cause the ODE solver to receive the MaxNFE error (Max Number of Function Evaluations exceeded). Hence, harming the overall performance.

In the following experiment, for each exponential value $\alpha$ in $\{0.0001, 0.001, 0.01, 0.1, 0.5\}$, we trained with 10 random weights initializations and increased the integration depth $T$ from 4 to 128 to observe the behavior the test accuracies. The results in Figure 2 suggest the performance of DeepGRAND under smaller $\alpha$ values are quite stable. This gives us a rough idea of what range of $\alpha$ values to use when we perform hyper-parameter tuning for DeepGRAND.

Table 3: The means and standard deviations of test accuracies of DeepGRAND-l, variants of GRAND, and other common GNNs experimented with different number of labelled nodes per class. Best results are written in bold (note: The results for GRAND++-l, GCN, GAT and GraphSage were imported from Thorpe et al. (2022)).

| Dataset | # labelled | GRAND-l | GRAND-nl | GRAND++-l | GCN | GAT | GraphSage | DeepGRAND-l (ours) |
|---|---|---|---|---|---|---|---|---|
| Cora | 20 | $82.86 \pm 1.12$ | $82.72 \pm 2.45$ | $82.95 \pm 1.37$ | $82.07 \pm 2.03$ | $79.92 \pm 2.28$ | $80.04 \pm 2.54$ | $\mathbf{84.20 \pm 0.71}$ |
| | 10 | $80.67 \pm 2.19$ | $80.51 \pm 1.11$ | $80.86 \pm 2.99$ | $78.82 \pm 5.38$ | $76.31 \pm 4.87$ | $75.04 \pm 5.03$ | $\mathbf{82.88 \pm 0.78}$ |
| | 5 | $77.09 \pm 3.05$ | $77.68 \pm 2.85$ | $77.80 \pm 4.46$ | $73.86 \pm 7.97$ | $71.04 \pm 5.74$ | $68.14 \pm 6.95$ | $\mathbf{80.85 \pm 1.08}$ |
| | 2 | $74.23 \pm 5.58$ | $69.44 \pm 4.27$ | $66.92 \pm 10.04$ | $60.85 \pm 14.01$ | $58.30 \pm 13.55$ | $53.96 \pm 12.18$ | $\mathbf{76.49 \pm 1.97}$ |
| | 1 | $57.93 \pm 8.09$ | $55.86 \pm 10.04$ | $54.94 \pm 16.09$ | $47.72 \pm 15.33$ | $47.86 \pm 15.38$ | $43.04 \pm 14.01$ | $\mathbf{70.15 \pm 3.25}$ |
| Citeseer | 20 | $71.74 \pm 2.94$ | $73.16 \pm 3.09$ | $73.53 \pm 3.31$ | $74.21 \pm 2.90$ | $73.22 \pm 2.90$ | $72.02 \pm 2.82$ | $\mathbf{74.67 \pm 0.78}$ |
| | 10 | $66.26 \pm 4.19$ | $67.84 \pm 3.96$ | $72.34 \pm 2.42$ | $72.18 \pm 3.47$ | $71.35 \pm 4.92$ | $68.90 \pm 5.08$ | $\mathbf{73.45 \pm 0.84}$ |
| | 5 | $69.00 \pm 3.74$ | $66.90 \pm 4.62$ | $70.03 \pm 3.63$ | $67.24 \pm 4.19$ | $67.37 \pm 5.08$ | $64.79 \pm 5.16$ | $\mathbf{71.97 \pm 1.03}$ |
| | 2 | $58.35 \pm 8.98$ | $56.35 \pm 5.54$ | $64.98 \pm 8.31$ | $58.06 \pm 9.76$ | $55.55 \pm 9.19$ | $54.39 \pm 11.37$ | $\mathbf{69.74 \pm 2.97}$ |
| | 1 | $49.65 \pm 8.67$ | $47.32 \pm 6.66$ | $\mathbf{58.95 \pm 9.59}$ | $48.94 \pm 10.24$ | $50.31 \pm 14.27$ | $48.81 \pm 11.45$ | $58.35 \pm 2.51$ |
| Pubmed | 20 | $78.42 \pm 0.46$ | $75.19 \pm 1.77$ | $79.16 \pm 1.37$ | $76.89 \pm 3.27$ | $75.55 \pm 4.11$ | $74.55 \pm 3.09$ | $\mathbf{79.50 \pm 0.64}$ |
| | 10 | $74.10 \pm 1.88$ | $74.18 \pm 1.99$ | $75.13 \pm 3.88$ | $72.59 \pm 3.19$ | $72.44 \pm 3.50$ | $70.74 \pm 3.11$ | $\mathbf{78.93 \pm 1.48}$ |
| | 5 | $71.05 \pm 1.87$ | $72.07 \pm 2.15$ | $71.99 \pm 1.91$ | $68.69 \pm 7.93$ | $68.54 \pm 5.75$ | $66.07 \pm 6.16$ | $\mathbf{77.09 \pm 1.12}$ |
| | 2 | $71.44 \pm 3.85$ | $65.50 \pm 9.49$ | $69.31 \pm 4.87$ | $60.45 \pm 16.20$ | $60.24 \pm 14.44$ | $58.97 \pm 12.65$ | $\mathbf{72.32 \pm 1.82}$ |
| | 1 | $62.41 \pm 7.59$ | $63.47 \pm 5.47$ | $65.94 \pm 4.87$ | $58.61 \pm 12.83$ | $58.84 \pm 12.81$ | $55.53 \pm 12.71$ | $\mathbf{70.03 \pm 1.84}$ |
| Computers | 20 | $84.04 \pm 0.98$ | $83.28 \pm 1.24$ | $85.73 \pm 0.50$ | $82.94 \pm 1.54$ | $80.05 \pm 1.81$ | $79.98 \pm 0.96$ | $\mathbf{87.12 \pm 0.45}$ |
| | 10 | $82.34 \pm 2.18$ | $81.27 \pm 3.02$ | $82.99 \pm 0.81$ | $82.53 \pm 0.74$ | $76.04 \pm 0.35$ | $74.66 \pm 1.29$ | $\mathbf{85.73 \pm 1.11}$ |
| | 5 | $78.69 \pm 0.79$ | $79.65 \pm 2.02$ | $\mathbf{82.64 \pm 0.56}$ | $82.47 \pm 0.97$ | $71.43 \pm 7.34$ | $64.83 \pm 1.62$ | $82.42 \pm 0.33$ |
| | 2 | $66.21 \pm 11.26$ | $65.11 \pm 8.31$ | $76.47 \pm 1.48$ | $\mathbf{76.90 \pm 1.49}$ | $65.07 \pm 8.86$ | $42.63 \pm 42.9$ | $76.57 \pm 1.44$ |
| | 1 | $49.80 \pm 14.97$ | $47.26 \pm 11.23$ | $67.65 \pm 0.37$ | $49.46 \pm 1.65$ | $37.14 \pm 7.81$ | $27.65 \pm 2.39$ | $\mathbf{69.33 \pm 2.71}$ |
| Photo | 20 | $93.22 \pm 0.44$ | $91.76 \pm 1.51$ | $\mathbf{93.55 \pm 0.38}$ | $91.95 \pm 0.11$ | $89.38 \pm 2.48$ | $91.29 \pm 0.67$ | $93.46 \pm 0.66$ |
| | 10 | $90.80 \pm 1.37$ | $89.02 \pm 2.54$ | $90.65 \pm 1.19$ | $90.41 \pm 0.35$ | $87.42 \pm 2.38$ | $84.38 \pm 1.75$ | $\mathbf{92.29 \pm 0.42}$ |
| | 5 | $88.06 \pm 2.59$ | $88.31 \pm 1.63$ | $88.33 \pm 1.21$ | $88.86 \pm 1.56$ | $83.01 \pm 3.64$ | $78.26 \pm 1.93$ | $\mathbf{90.45 \pm 0.99}$ |
| | 2 | $82.60 \pm 2.96$ | $80.61 \pm 4.43$ | $83.71 \pm 0.90$ | $83.61 \pm 0.71$ | $76.89 \pm 4.89$ | $51.93 \pm 4.21$ | $\mathbf{85.09 \pm 0.32}$ |
| | 1 | $75.05 \pm 5.44$ | $76.33 \pm 4.79$ | $83.12 \pm 0.78$ | $82.94 \pm 2.17$ | $73.58 \pm 8.15$ | $45.36 \pm 7.13$ | $\mathbf{83.27 \pm 1.88}$ |
| CoauthorCS | 20 | $90.99 \pm 0.56$ | $90.59 \pm 0.97$ | $90.80 \pm 0.34$ | $91.09 \pm 0.35$ | $79.95 \pm 2.88$ | $91.33 \pm 0.36$ | $\mathbf{91.66 \pm 0.59}$ |
| | 10 | $89.00 \pm 2.08$ | $\mathbf{89.95 \pm 0.68}$ | $86.94 \pm 0.46$ | $88.60 \pm 0.50$ | $74.71 \pm 3.35$ | $89.68 \pm 0.39$ | $89.80 \pm 0.70$ |
| | 5 | $84.19 \pm 3.59$ | $87.01 \pm 1.97$ | $84.83 \pm 0.84$ | $86.66 \pm 0.43$ | $71.65 \pm 4.53$ | $\mathbf{89.06 \pm 0.69}$ | $88.24 \pm 0.68$ |
| | 2 | $75.19 \pm 3.84$ | $76.66 \pm 6.85$ | $76.53 \pm 1.85$ | $\mathbf{83.61 \pm 1.49}$ | $63.12 \pm 6.09$ | $76.51 \pm 1.31$ | $82.08 \pm 3.39$ |
| | 1 | $56.58 \pm 8.44$ | $66.44 \pm 8.17$ | $60.30 \pm 1.50$ | $65.22 \pm 2.25$ | $51.13 \pm 5.24$ | $61.35 \pm 1.35$ | $\mathbf{71.14 \pm 1.66}$ |

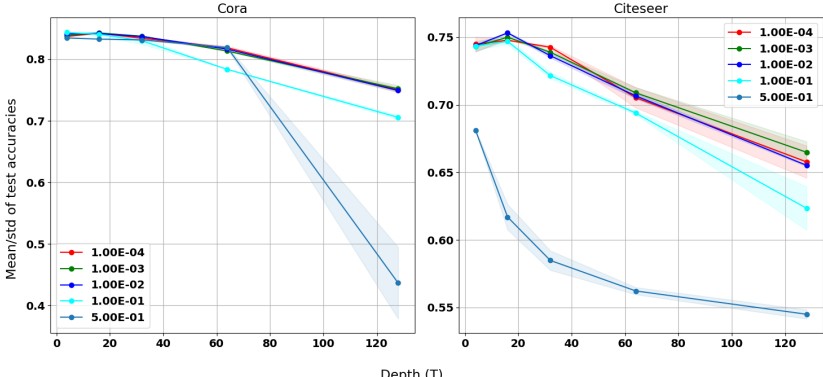

Figure 2: The change in test accuracies of DeepGRAND-l as depth value $T$ increases under different $\alpha$ values. **Left** - effect of $\alpha$ on the accuracies of Cora, **Right** - effect of $\alpha$ on the accuracies of Citeseer. Each $\alpha$ value is trained on 10 random seeds for both benchmarks.

## 6 CONCLUSION

We propose DeepGRAND, a class of continuous-depth graph neural networks that leverage a novel data-dependent scaling term and perturbation to the graph diffusivity to decrease the saturation rate of the underlying diffusion process, thus alleviating the over-smoothing issue. We also prove that the proposed method stabilize the learning of the model by controlling the norm of the node representation. We theoretically and empirically showed its advantage over GRAND and other popular GNNs in term of resiliency to over-smoothing and overall performance. DeepGRAND is a first-order system of ODEs. It is natural to explore an extension of DeepGRAND to a second-order system of ODEs and study the dynamics of this system. It is also interesting to leverage advanced methods in improving the Neural ODEs (Chen et al., 2018) to further develop DeepGRAND.

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

## A    TECHNICAL PROOFS

Consider the dynamic given by equation 3 and equation 4. We say that a matrix is postitive if and only if all of its entries are positive. Clearly, $\boldsymbol{A}$ is a postive right-stochastic matrix. We recall the well-known Perron-Frobenius theorem

**Theorem 1** (Perron-Frobenius). *Let $\boldsymbol{M}$ be a positive matrix. There is a positive real number $r$, called the Perron–Frobenius eigenvalue, such that $r$ is an eigenvalue of $M$ and any other eigenvalue $\lambda$ (possibly complex) in absolute value is strictly smaller than $r$ , $|\lambda| < r$. This eigenvalue is simple and its eigenspace is one-dimensional.*

*Proof of Proposition 1.* Since $\boldsymbol{A}$ is right-stochastic, its Perron-Frobenius eigenvalue is $\alpha_1 = 1$. Its eigenspace has the basis $\boldsymbol{u}_1 = \{1, 1, \ldots, 1\}$. Suppose $\{\alpha_1, \alpha_2, \ldots, \alpha_k\}$ is the complex spectrum of $\boldsymbol{A}$. That is, they are all complex eigenvalues of $\boldsymbol{A}$. The matrix $\boldsymbol{A} - \boldsymbol{I}$ has eigenvalues $\beta_i = \alpha_i - 1$ for all $i = \overline{1, k}$, so $\beta_1 = 0$ and $\operatorname{Re} \beta_i < 0$ for all $i = \overline{2, k}$. There exists a basis containing $\boldsymbol{u}_1$ of $\mathbb{C}^n$ comprising of generalized eigenvectors of $\boldsymbol{A}$. By rearranging the vectors if needed, the transition matrix $\boldsymbol{P}$ from the standard basis to this basis satisfies

$$\boldsymbol{P}^{-1}(\boldsymbol{A} - \boldsymbol{I})\boldsymbol{P} = \begin{pmatrix} 0 & 0 & \ldots & 0 \\ 0 & \boldsymbol{J}_1 & \ldots & 0 \\ \vdots & \vdots & \ddots & \vdots \\ 0 & 0 & \ldots & \boldsymbol{J}_m \end{pmatrix},$$

where each $\boldsymbol{J}_i$ is a Jordan block associated with some eigenvalue $\beta_i$. □

*Proof of Proposition 2.* Let $\boldsymbol{J} = \boldsymbol{P}^{-1}\boldsymbol{A}\boldsymbol{P}$ and $\boldsymbol{Z}(t) = \boldsymbol{P}^{-1}\boldsymbol{X}(t)$ as in Proposition 1, we can rewrite equation 4 as

$$\frac{\partial \boldsymbol{Z}}{\partial t}(t) = \boldsymbol{J}\boldsymbol{Z}(t),$$

The solution to this matrix differential equation is

$$\boldsymbol{Z}(t) = \exp(t\boldsymbol{J})\boldsymbol{Z}(0),$$

where

$$\exp(t\boldsymbol{J}) = \begin{pmatrix} 1 & 0 & 0 & \ldots & 0 \\ & \exp(t\boldsymbol{J}_1) & 0 & \ddots & \vdots \\ & & \exp(t\boldsymbol{J}_2) & \ddots & \vdots \\ & & & \ddots & \vdots \\ & & & & 0 \\ & & & & \exp(t\boldsymbol{J}_m) \end{pmatrix}.$$

Let $\overline{\boldsymbol{Z}}$ be the matrix with the same size as $\boldsymbol{Z}(0)$ obtained by setting every entry in all but the first row of $\boldsymbol{Z}(0)$ be equal to 0. We have

$$\boldsymbol{Z}(t) - \overline{\boldsymbol{Z}} = \boldsymbol{Z}(t) - \begin{pmatrix} 1 & 0 & \ldots & 0 \\ 0 & 0 & \ldots & 0 \\ \vdots & \vdots & \ddots & \vdots \\ 0 & 0 & \ldots & 0 \end{pmatrix} \boldsymbol{Z}(0) = \begin{pmatrix} 0 & 0 & 0 & \ldots & 0 \\ & \exp(t\boldsymbol{J}_1) & 0 & \ddots & \vdots \\ & & \exp(t\boldsymbol{J}_2) & \ddots & \vdots \\ & & & \ddots & \vdots \\ & & & & 0 \\ & & & & \exp(t\boldsymbol{J}_m) \end{pmatrix} \boldsymbol{Z}(0).$$

Denote the size of each Jordan block $\boldsymbol{J}_i$ as $n_i$, we have well-known equality

$$\exp(t\boldsymbol{J}_i) = e^{\beta_i t}\begin{pmatrix} 1 & t & \frac{t^2}{2} & \cdots & \frac{t^{n_i-1}}{(n_i-1)!} \\ & 1 & t & \ddots & \vdots \\ & & 1 & \ddots & \vdots \\ & & & \ddots & \vdots \\ & & & & t \\ & & & & 1 \end{pmatrix}.$$

Hence, every non-zero entry of $\boldsymbol{Z}(t) - \overline{\boldsymbol{Z}}$ has the form $e^{\beta_i t}P(t)$ for some $\beta_i < 0$ and polynomial $P$. Each entry can thus be bounded above in norm by an exponential term of the form $c_1 e^{-c_2 t}$ for some $c_1, c_2 > 0$. A simple calculation shows that $\|\boldsymbol{X}(t) - \boldsymbol{P}\overline{\boldsymbol{Z}}\|_\infty = \|\boldsymbol{P}(\boldsymbol{Z}(t) - \overline{\boldsymbol{Z}})\|_\infty$ would also be bounded by an exponential term. That is, there exists some $C_1, C_2 > 0$ such that

$$\|\boldsymbol{X}(t) - \boldsymbol{P}\overline{\boldsymbol{Z}}\|_\infty \le C_1 e^{-C_2 t}. \tag{12}$$

Recall that the first column of $\boldsymbol{P}$ is $\boldsymbol{u}^\top = (1, 1, \ldots, 1)^\top$, so that

$$\boldsymbol{P}\overline{\boldsymbol{Z}} = \begin{pmatrix} \boldsymbol{Z}(0)_{1,1} & \boldsymbol{Z}(0)_{1,2} & \ldots & \boldsymbol{Z}(0)_{1,d} \\ \boldsymbol{Z}(0)_{1,1} & \boldsymbol{Z}(0)_{1,2} & \ldots & \boldsymbol{Z}(0)_{1,d} \\ \vdots & \vdots & \ddots & \vdots \\ \boldsymbol{Z}(0)_{1,1} & \boldsymbol{Z}(0)_{1,2} & \ldots & \boldsymbol{Z}(0)_{1,d} \end{pmatrix}.$$

Let $\boldsymbol{v} = (\boldsymbol{Z}(0)_{1,1}, \boldsymbol{Z}(0)_{1,2}, \ldots, \boldsymbol{Z}(0)_{1,d})^\top$ and $\boldsymbol{V} = (\boldsymbol{v}, \boldsymbol{v}, \ldots, \boldsymbol{v})^\top$. We can rewrite equation 12 as

$$\|\boldsymbol{X}(t) - \boldsymbol{V}\|_\infty \le C_1 e^{-C_2 t}. \tag{13}$$

The claim that $\boldsymbol{v} \in \mathbb{R}^d$ follows from the fact that it is the limit of real-valued vectors with regards to the $\|\cdot\|_\infty$ norm. $\qquad\square$

*Proof of Proposition 3.* Let $\boldsymbol{Y}_i(t) = \|\boldsymbol{X}_i(t)\|^4$, we have

$$\begin{aligned} \boldsymbol{Y}_i'(t) &= 4\|\boldsymbol{X}_i(t)\|^2\langle \boldsymbol{X}_i'(t), \boldsymbol{X}_i(t)\rangle \\ &= 4\|\boldsymbol{X}_i(t)\|^2\langle (\boldsymbol{A} - (1+\epsilon)\boldsymbol{I})\boldsymbol{X}_i(t)\|\boldsymbol{X}_i\|^\alpha, \boldsymbol{X}_i(t)\rangle \\ &= 4\|\boldsymbol{X}_i(t)\|^{2+\alpha}\langle (\boldsymbol{A} - (1+\epsilon)\boldsymbol{I})\boldsymbol{X}_i(t), \boldsymbol{X}_i(t)\rangle \\ &= 4\|\boldsymbol{X}_i(t)\|^{2+\alpha}\langle \boldsymbol{A}\boldsymbol{X}_i(t), \boldsymbol{X}_i(t)\rangle - 4(1+\epsilon)\|\boldsymbol{X}_i(t)\|^{4+\alpha} \end{aligned}$$

Since $\boldsymbol{A}$ is a right-stochastic and radial matrix, we have

$$\langle \boldsymbol{A}\boldsymbol{X}_i(t), \boldsymbol{X}_i(t)\rangle \le \|\boldsymbol{A}\|\|\boldsymbol{X}_i(t)\|^2 = \|\boldsymbol{X}_i(t)\|^2,$$

and

$$-\langle \boldsymbol{A}\boldsymbol{X}_i(t), \boldsymbol{X}_i(t)\rangle \le \|\boldsymbol{A}\|\|\boldsymbol{X}_i(t)\|^2 = \|\boldsymbol{X}_i(t)\|^2,$$

so that

$$-\|\boldsymbol{X}_i(t)\|^2 \le \langle \boldsymbol{A}\boldsymbol{X}_i(t), \boldsymbol{X}_i(t)\rangle \le \|\boldsymbol{X}_i(t)\|^2.$$

Hence, we deduce that

$$4\|\boldsymbol{X}_i(t)\|^{4+\alpha}(-2-\epsilon) \le \boldsymbol{Y}_i'(t) \le 4\|\boldsymbol{X}_i(t)\|^{4+\alpha}(-\epsilon). \tag{14}$$

Multiply both sides of equation 14 with $-\frac{\alpha}{4}\|\boldsymbol{X}_i(t)\|^{-4-\alpha} = -\frac{\alpha}{4}\boldsymbol{Y}_i(t)^{-1-\alpha/4}$, and by noting that $-\frac{\alpha}{4}\boldsymbol{Y}_i'\boldsymbol{Y}_i^{-1-\alpha/4} = (\boldsymbol{Y}_i^{-\alpha/4})'$, we have

$$\alpha(2+\epsilon) \ge (\boldsymbol{Y}_i^{-\alpha/4})' \ge \alpha\epsilon.$$

Integrate from $0$ to $T$ and rearranging the appropriate terms, we get

$$(2+\epsilon)\alpha T + \boldsymbol{Y}_i(0)^{-\alpha/4} \ge \boldsymbol{Y}_i(T)^{-\alpha/4} \ge \epsilon\alpha T + \boldsymbol{Y}_i(0)^{-\alpha/4}.$$

Finally, by noting that $\boldsymbol{Y}_i^{\frac{-\alpha}{4}} = \|\boldsymbol{X}_i\|^{-\alpha}$, we easily get the bound equation 10

$$\left((2+\epsilon)\alpha T + \|\boldsymbol{X}_i(0)\|^{-\alpha}\right)^{\frac{-1}{\alpha}} \le \|\boldsymbol{X}_i(T)\| \le \left(\epsilon\alpha T + \|\boldsymbol{X}_i(0)\|^{-\alpha}\right)^{\frac{-1}{\alpha}}.$$

$\qquad\square$

## B    PHYSICAL INTERPRETATION OF GRAND

Informally, diffusion describes the process by which a diffusing material is transported from a region of higher to lower density through random microscopic motions. A natural example is the process of heat diffusion, which occurs when a hot object touches a cold object. Heat will diffuse between them until both objects are of the same temperature.

GRAND approaches deep learning on graphs as a continuous diffusion process and treat GNNs as discretisations of an underlying PDE. In doing so, the dynamic in GRAND is the same as that in (heat) diffusion problems. We use this viewpoint to consider the over-smoothing issue.

Consider each node in a given graph as a point in space containing some amount of thermal energy. Each pair of points is connected through a heat pipe. The thermal conductivity (also known as the diffusivity) of each pipe is specific to each pair of points. A thermal conductivity of $0$ represents two points that are not connected in the graph. The exact formulation of graph diffusion up to equation 4 represents a closed system. That is, the total amount of energy in all nodes is invariant of time.

In node classification problems, nodes within the same class are often hypothesized as sharing strong connection with each other. This is known as the homophily assumption, i.e. 'like attracts like'. For example, friends are more likely to share an interest, and papers from the same research area tends to cite each other. Homophily is a key principle of many real-world networks, and its effect on GNNs has gained traction as a research direction (Zhu et al., 2020; Yan et al., 2021; Chen et al., 2022; Luan et al., 2022; Ma et al., 2022).

In the context of GRAND, we can assume nodes within the same class as sharing a highly conductive heat pipe. As such, thermal energy is transferred effectively between them, quickly bridging any gap in temperature. This allows for rapid clustering of nodes into classes of different thermal energy levels, which can then be passed into a fully connected layer to perform classification.

This interpretation gives a surprising intuitive explanation to the occurrence of the over-smoothing issue in GRAND. If the dynamic carries on for too long, the energy level of every node will exponentially converge to the average thermal energy, given that the graph is sufficiently connected. Hence, we argue that GRAND (and more generally, any GNN based purely on diffusion) is inherently prone to suffer from over-smoothing.

## C    ON NEURAL ODES

Traditional neural networks such as residual neural networks, normalizing flows, recurrent-neural-networks (RNNs) learn complicated mappings via the composition of multiple transformations to the hidden states. For example, the residual network updates the future hidden state using the following equation:

$$\boldsymbol{h}_{t+1} = \boldsymbol{h}_t + f(\boldsymbol{h}_t, t, \theta) \tag{15}$$

Where $t \in \{1, \ldots, T\}$ is the layer index of the neural network and $\boldsymbol{h}_t$ is the hidden state at layer $t$. This iterative update rule can be seen as the Euler discretisation of a continuous transformation.

Neural ODEs (Chen et al., 2018) are a class of continuous-depth (or continuous-time) neural networks where the transformation step $t$ is infinitesimally small. Specifically, the hidden state $\boldsymbol{h}_t$ is parameterized using a continuous dynamics with respect to time $t$:

$$\frac{d\boldsymbol{h}_t}{dt} = f(\boldsymbol{h}_t, t, \theta) \tag{16}$$

Where $f(\boldsymbol{h}_t, t, \theta)$ is specified by a neural network parameterized by $\theta$. Starting from the initial state $\boldsymbol{h}(0)$, Neural ODEs learn the final representation $\boldsymbol{h}(T)$ by solving 16 using a numerical integrator (often with an adaptive step-size solver or an adaptive solver for short) given an error tolerance. Integrating 16 from $0$ to $T$ in a single forward pass requires the adaptive solver to evaluate $f(\boldsymbol{h}_t, t, \theta)$ at multiple time-steps. The computational complexity of the forward pass is determined by the number of function evaluations.

The only problem with updating the hidden state by solving 16 numerically is that the black-box ODE solver is not a differentiable operation. Therefore, the usual back-propagation method for optimizing traditional neural networks does not work for Neural ODEs.

The adjoint sensitivity method (or adjoint method) is a memory-efficient alternative of the traditional back-propagation method for training Neural ODEs. We denote $\boldsymbol{h}(T)$ as the prediction of Neural ODEs and the loss between $\boldsymbol{h}(T)$ and the ground truth is $\mathcal{L}$. Then, we define the adjoint state as $\boldsymbol{a}(t) = \partial \mathcal{L} / \partial \boldsymbol{h}(t)$, we have:

$$\frac{d\mathcal{L}}{d\theta} = \int_0^T \boldsymbol{a}(t)^T \frac{\partial f(\boldsymbol{h}(t), t, \theta)}{\partial \theta} dt \tag{17}$$

Where the adjoint state $\boldsymbol{a}(t)$ satisfies the following dynamics:

$$\frac{d\boldsymbol{a}(t)}{dt} = -\boldsymbol{a}(t)^T \frac{\partial f(\boldsymbol{h}(t), t, \theta)}{\partial \boldsymbol{h}(t)} \tag{18}$$

Since the adjoint state in 18 can be solved numerically using an ODE solver, the gradient of the loss function $\mathcal{L}$ with respect to $\theta$ in 17 can be evaluated. The computational complexity of the backward pass is determined by the number of function evaluations used by the ODE solver to solve for the adjoint state.

# D  EULER DISCRETIZATION OF GRAND ALSO SUFFERS FROM OVER-SMOOTHING

Recall that the forward Euler discretization of the GRAND dynamic was given by Thorpe et al. (2022) as

$$\boldsymbol{X}(k\delta_t) = \boldsymbol{X}((k-1)\delta_t) + \delta_t(\boldsymbol{A} - \boldsymbol{I})\boldsymbol{X}((k-1)\delta_t), \tag{19}$$

where $1 > \delta_t > 0$ is the fixed step size, $k = 1, 2, \ldots, K$ denotes the layers from 1 to $K$ and $\boldsymbol{X}_k := \boldsymbol{X}(k\delta_t)$ is the node feature at the $k$-th layer.

We mirror our analysis as in Section 3 almost completely. Some calculations will be omitted for clarity. First, we give an analogous definition to 1 for discrete GNNs.

**Definition 2.** *Let $\boldsymbol{X}_k \in \mathbb{R}^{n \times d}$ denote the feature representation at the $k$-th layer of some discrete GNN dynamic. $(\boldsymbol{X}_k)$ is said to experience over-smoothing if there exists a vector $\boldsymbol{v} \in \mathbb{R}^d$ and constants $C_1, C_2 > 0$ such that for $\boldsymbol{V} = (\boldsymbol{v}, \boldsymbol{v}, \ldots, \boldsymbol{v})^\top$*

$$\|\boldsymbol{X}_k - \boldsymbol{V}\|_\infty \leq C_1 e^{-C_2 k}. \tag{20}$$

Utilising Proposition 1, we can show that

**Proposition 4.** *With the dynamic given by equation 5 and equation 19, $(\boldsymbol{X}_k)$ experiences over-smoothing.*

*Proof.* Let $\boldsymbol{J} = \boldsymbol{P}^{-1}\boldsymbol{A}\boldsymbol{P}$ and $\boldsymbol{Z}_k = \boldsymbol{P}^{-1}\boldsymbol{X}_k$ as in Proposition 1, we can rewrite equation 19 as

$$\boldsymbol{Z}_k = \boldsymbol{Z}_{k-1} + \delta_t(\boldsymbol{J} - \boldsymbol{I})\boldsymbol{Z}_{k-1} = ((1-\delta_t)\boldsymbol{I} + \delta_t\boldsymbol{J})\,\boldsymbol{Z}_{k-1} = ((1-\delta_t)\boldsymbol{I} + \delta_t\boldsymbol{J})^k\,\boldsymbol{Z}_0, \tag{21}$$

where

$$((1-\delta_t)\boldsymbol{I} + \delta_t\boldsymbol{J})^k = \begin{pmatrix} 1 & 0 & \cdots & & 0 \\ & ((1-\delta_t)\boldsymbol{I} + \delta_t\boldsymbol{J}_1)^k & \ddots & & \vdots \\ & & \ddots & & \vdots \\ & & & & 0 \\ & & & & ((1-\delta_t)\boldsymbol{I} + \delta_m\boldsymbol{J})^k \end{pmatrix}.$$

Let $\overline{\boldsymbol{Z}}$ be the matrix with the same size as $\boldsymbol{Z}_0$ obtained by setting every entry in all but the first row of $\boldsymbol{Z}_0$ be equal to 0. We have

$$\boldsymbol{Z}_k - \overline{\boldsymbol{Z}} = \begin{pmatrix} 0 & 0 & \cdots & & 0 \\ & ((1-\delta_t)\boldsymbol{I} + \delta_t\boldsymbol{J}_1)^k & \ddots & & \vdots \\ & & \ddots & & \vdots \\ & & & & 0 \\ & & & & ((1-\delta_t)\boldsymbol{I} + \delta_m\boldsymbol{J})^k \end{pmatrix} \boldsymbol{Z}_0.$$

Each block $(1-\delta_t)\boldsymbol{I}+\delta_t\boldsymbol{J}_1$ has spectral radius lesser than 1. Hence, every non-zero entry of $\boldsymbol{Z}_k-\overline{\boldsymbol{Z}}$ can be bounded above in norm by an exponential term of the form $c_1 e^{-c_2 t}$ for some $c_1, c_2 > 0$. We can deduce that there exists some $C_1, C_2 > 0$ such that

$$\|\boldsymbol{X}_k - \boldsymbol{P}\overline{\boldsymbol{Z}}\|_\infty \le C_1 e^{-C_2 k}. \tag{22}$$

Recall that the first column of $\boldsymbol{P}$ is $\boldsymbol{u}^\top = (1,1,\ldots,1)^\top$. Set $\boldsymbol{v}$ to be the first row of $\boldsymbol{Z}_0$. We can rewrite equation 22 as

$$\|\boldsymbol{X}_k - \boldsymbol{V}\|_\infty \le C_1 e^{-C_2 k}.$$

The claim that $\boldsymbol{v} \in \mathbb{R}^d$ follows from the fact that it is the limit of real-valued vectors with regards to the $\|\cdot\|_\infty$ norm. □

# E   DEEPGRAND PERFORMANCE ON DIFFERENT NUMBERS OF LABELLED NODES PER CLASS

In this section, we provide additional experiment results to complement Table 3. Apart from the test accuracies of DeepGRAND ran on optimal $T$ values, we add an additional column showing DeepGRAND's results when trained with identical $T$ values as were used in the GRAND paper (Chamberlain et al. (2021b)). The experiments suggest that with the same $T$ values, DeepGRAND already outperforms GRAND on most of the benchmarks with limited number of labeled nodes. We observe that with optimal $T$ values, DeepGRAND has much less test accuracy variances compared to all other designs.

Table 4: The means and standard deviations of test accuracies of DeepGRAND-l and other variants of GRAND experimented with different number of labelled nodes per class. Best results are written in bold (Note: The results for GRAND++-l were imported from Thorpe et al. (2022)).

| Dataset | # labelled | GRAND-l | GRAND-nl | GRAND++-l | DeepGRAND-l (GRAND setting) | DeepGRAND-l (optimal T) |
|---|---|---|---|---|---|---|
| Cora | 20 | $82.86 \pm 1.12$ | $82.72 \pm 2.45$ | $82.95 \pm 1.37$ | $83.87 \pm 0.49$ | $\mathbf{84.20 \pm 0.71}$ |
| | 10 | $80.67 \pm 2.19$ | $80.51 \pm 1.11$ | $80.86 \pm 2.99$ | $82.47 \pm 1.11$ | $\mathbf{82.88 \pm 0.78}$ |
| | 5 | $77.09 \pm 3.05$ | $77.68 \pm 2.85$ | $77.80 \pm 4.46$ | $79.00 \pm 1.87$ | $\mathbf{80.85 \pm 1.08}$ |
| | 2 | $74.23 \pm 5.58$ | $69.44 \pm 4.27$ | $66.92 \pm 10.04$ | $67.57 \pm 8.02$ | $\mathbf{76.49 \pm 1.97}$ |
| | 1 | $57.93 \pm 8.09$ | $55.86 \pm 10.04$ | $54.94 \pm 16.09$ | $63.79 \pm 7.14$ | $\mathbf{70.15 \pm 3.25}$ |
| Citeseer | 20 | $71.74 \pm 2.94$ | $73.16 \pm 3.09$ | $73.53 \pm 3.31$ | $73.64 \pm 1.73$ | $\mathbf{74.67 \pm 0.78}$ |
| | 10 | $66.26 \pm 4.19$ | $67.84 \pm 3.96$ | $72.34 \pm 2.42$ | $72.87 \pm 2.29$ | $\mathbf{73.45 \pm 0.84}$ |
| | 5 | $69.00 \pm 3.74$ | $66.90 \pm 4.62$ | $70.03 \pm 3.63$ | $70.84 \pm 3.04$ | $\mathbf{71.97 \pm 1.03}$ |
| | 2 | $58.35 \pm 8.98$ | $56.35 \pm 5.54$ | $64.98 \pm 8.31$ | $63.26 \pm 3.89$ | $\mathbf{69.74 \pm 2.97}$ |
| | 1 | $49.65 \pm 8.67$ | $47.32 \pm 6.66$ | $\mathbf{58.95 \pm 9.59}$ | $53.06 \pm 8.49$ | $58.35 \pm 2.51$ |
| Pubmed | 20 | $78.42 \pm 0.46$ | $75.19 \pm 1.77$ | $79.16 \pm 1.37$ | $79.23 \pm 1.23$ | $\mathbf{79.50 \pm 0.64}$ |
| | 10 | $74.10 \pm 1.88$ | $74.18 \pm 1.99$ | $75.13 \pm 3.88$ | $76.95 \pm 3.24$ | $\mathbf{78.93 \pm 1.48}$ |
| | 5 | $71.05 \pm 1.87$ | $72.07 \pm 2.15$ | $71.99 \pm 1.91$ | $73.37 \pm 2.26$ | $\mathbf{77.09 \pm 1.12}$ |
| | 2 | $71.44 \pm 3.85$ | $65.50 \pm 9.49$ | $69.31 \pm 4.87$ | $70.75 \pm 3.86$ | $\mathbf{72.32 \pm 1.82}$ |
| | 1 | $62.41 \pm 7.59$ | $63.47 \pm 5.47$ | $65.94 \pm 4.87$ | $64.96 \pm 7.08$ | $\mathbf{70.03 \pm 1.84}$ |
| Computers | 20 | $84.04 \pm 0.98$ | $83.28 \pm 1.24$ | $85.73 \pm 0.50$ | $87.27 \pm 1.45$ | $\mathbf{87.12 \pm 0.45}$ |
| | 10 | $82.34 \pm 2.18$ | $81.27 \pm 3.02$ | $82.99 \pm 0.81$ | $83.79 \pm 1.45$ | $\mathbf{85.73 \pm 1.11}$ |
| | 5 | $78.69 \pm 0.79$ | $79.65 \pm 2.02$ | $\mathbf{82.64 \pm 0.56}$ | $82.06 \pm 2.02$ | $82.42 \pm 0.33$ |
| | 2 | $66.21 \pm 11.26$ | $65.11 \pm 8.31$ | $76.47 \pm 1.48$ | $75.26 \pm 2.20$ | $\mathbf{76.57 \pm 1.44}$ |
| | 1 | $49.80 \pm 14.97$ | $47.26 \pm 11.23$ | $67.65 \pm 0.37$ | $65.35 \pm 6.65$ | $\mathbf{69.33 \pm 2.71}$ |
| Photo | 20 | $93.22 \pm 0.44$ | $91.76 \pm 1.51$ | $\mathbf{93.55 \pm 0.38}$ | $93.52 \pm 0.40$ | $93.46 \pm 0.66$ |
| | 10 | $90.80 \pm 1.37$ | $89.02 \pm 2.54$ | $90.65 \pm 1.19$ | $90.59 \pm 2.27$ | $\mathbf{92.29 \pm 0.42}$ |
| | 5 | $88.06 \pm 2.59$ | $88.31 \pm 1.63$ | $88.33 \pm 1.21$ | $87.75 \pm 1.59$ | $\mathbf{90.45 \pm 0.99}$ |
| | 2 | $82.60 \pm 2.96$ | $80.61 \pm 4.43$ | $83.71 \pm 0.90$ | $84.59 \pm 1.80$ | $\mathbf{85.09 \pm 0.32}$ |
| | 1 | $75.05 \pm 5.44$ | $76.33 \pm 4.79$ | $83.12 \pm 0.78$ | $73.50 \pm 5.06$ | $\mathbf{83.27 \pm 1.88}$ |
| CoauthorCS | 20 | $90.99 \pm 0.56$ | $90.59 \pm 0.97$ | $90.80 \pm 0.34$ | $91.53 \pm 0.33$ | $\mathbf{91.66 \pm 0.59}$ |
| | 10 | $89.00 \pm 2.08$ | $\mathbf{89.95 \pm 0.68}$ | $86.94 \pm 0.46$ | $89.39 \pm 0.89$ | $89.80 \pm 0.70$ |
| | 5 | $84.19 \pm 3.59$ | $87.01 \pm 1.97$ | $84.83 \pm 0.84$ | $86.05 \pm 4.64$ | $\mathbf{88.24 \pm 0.68}$ |
| | 2 | $75.19 \pm 3.84$ | $76.66 \pm 6.85$ | $76.53 \pm 1.85$ | $79.25 \pm 3.89$ | $\mathbf{82.08 \pm 3.39}$ |
| | 1 | $56.58 \pm 8.44$ | $66.44 \pm 8.17$ | $60.30 \pm 1.50$ | $65.15 \pm 6.34$ | $\mathbf{71.14 \pm 1.66}$ |

Table 5: $T$ values of DeepGRAND used for different benchmarks.

| | Cora | Citeseer | Pubmed | CoauthorCS | Computers | Photo |
|---|---|---|---|---|---|---|
| **DeepGRAND (GRAND setting)** | 18.29 | 7.87 | 12.94 | 3.12 | 3.24 | 3.58 |
| **DeepGRAND (optimal T)** | 20.29 | 9.87 | 14.94 | 6.58 | 6.25 | 8.58 |

