# OpenReview forum: "DeepGRAND: Deep Graph Neural Diffusion"
_ICLR.cc/2023/Conference — Submitted to ICLR 2023_

### Official Review · Reviewer_3KbW · 2022-10-23

**Confidence:** 4
**Clarity, Quality, Novelty And Reproducibility:** The clarity, general quality and nove…
**Correctness:** 3
**Technical Novelty And Significance:** 3
**Empirical Novelty And Significance:** 2
**Recommendation:** 5

**Strength And Weaknesses:**

Strength:
1.	Convincible theoretical analysis on the diffusion process, the over-smoothing, and the bound of the feature.
2.	Fully designed experiments show the performance of the proposed method systematically.
3.	Well organized paper and logical writing.

Weaknesses:
1.	Lack of the explanation about the importance and the necessity to design deep GNN models . In this paper, the author tries to address the issue of over-smoothing and build deeper GNN models. However, there is no explanation about why should we build a deep GNN model. For CNN, it could be built for thousands of layers with significant improvement of the performance. While for GNN, the performance decreases with the increase of the depth (shown in Figure 1). Since the deeper GNN model does not show the significant improvement and consumes more computational resource, the reviewer wonders the explanation of the importance and the necessity to design deep models.
2.	The experimental results are not significantly improved compared with GRAND. For example, GRAND++-l on Cora with T=128 in Table 1, on Computers with T=16,32 in Table 2. Since the author claims that GRAND suffers from the over-smoothing issue while DeepGRAND significantly mitigates such issue, how to explain the differences between the theoretical and practical results, why GRAND performs better when T is larger? Besides, in Table 3, DeepGRAND could not achieve the best performance with 1/2 labeled on Citeseer, Pubmed, Computers and CoauthorCS dataset, which could not support the argument that DeepGRAND is more resilient under limited labeled training data.
3.	Insufficient ablation study on \alpha. \alpha is only set to 1e-4, 1e-1, 5e-1 in section 5.4 with a large gap between 1e-4 and 1e-1. The author is recommended to provide more values of \alpha, at least 1e-2 and 1e-3.
4.	Minor issues. The x label of Figure 2, Depth (T) rather than Time (T).


**Summary Of The Paper:**

In this paper, the authors propose the Deep Graph Neural Diffusion method, which ensures the over-smoothing issue and guarantees stability of the model in theoretical. The experimental results also show the improvement of the proposed method compared with the baseline methods.

**Summary Of The Review:**

This paper provides convincible theoretical analysis on the diffusion process, the over-smoothing, and the bound of the feature. However, the reviewer still has some concerns, listed below.
1.	Lack of the explanation about the importance and the necessity to design deep GNN models . In this paper, the author tries to address the issue of over-smoothing and build deeper GNN models. However, there is no explanation about why should we build a deep GNN model. For CNN, it could be built for thousands of layers with significant improvement of the performance. While for GNN, the performance decreases with the increase of the depth (shown in Figure 1). Since the deeper GNN model does not show the significant improvement and consumes more computational resource, the reviewer wonders the explanation of the importance and the necessity to design deep models.
2.	The experimental results are not significantly improved compared with GRAND. For example, GRAND++-l on Cora with T=128 in Table 1, on Computers with T=16,32 in Table 2. Since the author claims that GRAND suffers from the over-smoothing issue while DeepGRAND significantly mitigates such issue, how to explain the differences between the theoretical and practical results, why GRAND performs better when T is larger? Besides, in Table 3, DeepGRAND could not achieve the best performance with 1/2 labeled on Citeseer, Pubmed, Computers and CoauthorCS dataset, which could not support the argument that DeepGRAND is more resilient under limited labeled training data.
3.	Insufficient ablation study on \alpha. \alpha is only set to 1e-4, 1e-1, 5e-1 in section 5.4 with a large gap between 1e-4 and 1e-1. The author is recommended to provide more values of \alpha, at least 1e-2 and 1e-3, etc.
4.	Minor issue. The x label of Figure 2, Depth (T) rather than Time (T).

---

> ### Author Response · Authors · 2022-11-18
> **Response to Reviewer 3KbW (1)**
>
> Thank you for your thoughtful review and valuable feedback. Below we address your concerns.
>
> -----
> **Q1. Lack of the explanation about the importance and the necessity to design deep GNN models . In this paper, the author tries to address the issue of over-smoothing and build deeper GNN models. However, there is no explanation about why should we build a deep GNN model. For CNN, it could be built for thousands of layers with significant improvement of the performance. While for GNN, the performance decreases with the increase of the depth (shown in Figure 1). Since the deeper GNN model does not show the significant improvement and consumes more computational resources, the reviewer wonders the explanation of the importance and the necessity to design deep models**
>
> **Reply:** As the reviewer pointed out, CNN could be built for thousands of layers with significant improvement in performance. While for GNN, the performance decreases with the increase in depth. Over-smoothing is the cause of this performance degradation in GNNs as explained in Section 3 of our paper and in [1,2,3]. The goal of our paper is to propose a new method that allows GNNs to overcome the over-smoothing issue via scaling and perturbing the graph diffusivity. After the over-smoothing issue is mitigated, the GNNs can gain more representation power with depth like the CNNs [4,5,6] to achieve better accuracy on large graphs which require the interaction of far-away nodes. The importance of depth in GNNs is also discussed in [7,8,9].
>
> **References**
>
> [1] Kenta Oono and Taiji Suzuki. Graph neural networks exponentially lose expressive power for node classification. In International Conference on Learning Representations, 2020.
>
> [2] Chen Cai and Yusu Wang.A note on over-smoothing for graph neural networks, 2020.
>
> [3] T. Konstantin Rusch, Benjamin P. Chamberlain, James Rowbottom, Siddhartha Mishra, and Michael M. Bronstein. Graph-coupled oscillator networks. In International Conference on Machine Learning, 2022.
>
> [4] Li, Guohao, Matthias Muller, Ali Thabet, and Bernard Ghanem. Deepgcns: Can gcns go as deep as cnns?. In Proceedings of the IEEE/CVF international conference on computer vision, pp. 9267-9276. 2019.
>
> [5] Li, Guohao, Chenxin Xiong, Ali Thabet, and Bernard Ghanem. Deepergcn: All you need to train deeper gcns. arXiv preprint arXiv:2006.07739 (2020).
>
> [6] Li, Guohao, Matthias Müller, Bernard Ghanem, and Vladlen Koltun. Training graph neural networks with 1000 layers. In International conference on machine learning, pp. 6437-6449. PMLR, 2021.
>
> [7] Dehmamy, Nima, Albert-László Barabási, and Rose Yu. Understanding the representation power of graph neural networks in learning graph topology. Advances in Neural Information Processing Systems 32 (2019).
>
> [8] Loukas, Andreas. What graph neural networks cannot learn: depth vs width. arXiv preprint arXiv:1907.03199 (2019). In International Conference on Learning Representations, 2020.
>
> [9] Zeng, Hanqing, Muhan Zhang, Yinglong Xia, Ajitesh Srivastava, Andrey Malevich, Rajgopal Kannan, Viktor Prasanna, Long Jin, and Ren Chen. Decoupling the depth and scope of graph neural networks. Advances in Neural Information Processing Systems 34 (2021): 19665-19679.

---

> > ### Author Response · Authors · 2022-11-18
> > **Response to Reviewer 3KbW (2)**
> >
> > **Q2. The experimental results are not significantly improved compared with GRAND. For example, GRAND++-l on Cora with T=128 in Table 1, on Computers with T=16,32 in Table 2. Since the author claims that GRAND suffers from the over-smoothing issue while DeepGRAND significantly mitigates such issue, how to explain the differences between the theoretical and practical results, why GRAND performs better when T is larger? Besides, in Table 3, DeepGRAND could not achieve the best performance with 1/2 labeled on Citeseer, Pubmed, Computers and CoauthorCS dataset, which could not support the argument that DeepGRAND is more resilient under limited labeled training data.**
> >
> > **Reply:** We believe there is confusion between GRAND and GRAND++-l. Please allow us to clear this confusion by clarifying the difference between GRAND and GRAND++-l and the empirical advantages of our DeepGRAND over these two methods. GRANT++-l modifies GRAND by adding source terms to the original dynamics to keep the diffusion in GRAND from saturating [1]. Our DeepGRAND outperforms GRAND in all settings in Tables 1, 2, and 3 in the main text and in Table 4 in the Appendix. Our DeepGRAND and GRAND++-l are two orthogonal efforts that try to mitigate over-smoothing, i.e. feature saturation, in GRAND. Our DeepGRAND outperforms GRAND++-l  in most of the depth settings in Tables 1 and 2. However, we observe that GRAND++-l  becomes more stiff and unstable when the $T$ values are high, i.e., greater depth,  causing the ODE solver fails to converge. For example, GRAND++-l fails on Pubmed with $T=64$, $T=128$, on Citeseer with $T=128$, and on CoauthorCS with $T=32$.
> >
> > In Table 3 in our previous manuscript, the experimental results show that on Cora, Citeseer, and Pubmed, DeepGRAND achieves better performance than most of the baseline methods while being on par with the other when the number of labeled nodes are 1, 2, 5, 10, and 20. However, for the later 3 benchmarks (CoauthorCS, Computers, Photo), DeepGRAND’s performance is worse than the baseline methods. This is because, in this experiment, we reused the depth $T$ tuned for the linear case of GRAND. Therefore, we have provided additional experimental results for this ablation study with $T$ values specifically fine-tuned for DeepGRAND in Table 3 in the revised manuscript.  In Table 4 in Appendix E of our revision, we compare the results of GRAND-l, GRAND-nl, GRAND++-l, the old experiments of DeepGRAND, and the fine-tuned version of DeepGRAND. We provided the specific list of $T$ values for DeepGRAND for each benchmark in Table 5. The new experimental results suggest that DeepGRAND significantly outperforms other methods including GRAND++-l across all benchmarks under different label rates. Furthermore, the standard deviations of the test accuracies are remarkably lower than those of the baseline methods, indicating that DeepGRAND is more stable than the baselines in a low label rate regime.
> >
> > **References**
> >
> > [1] Matthew Thorpe, Tan Minh Nguyen, Hedi Xia, Thomas Strohmer, Andrea Bertozzi, Stanley Osher, and Bao Wang. GRAND++: Graph neural diffusion with a source term. In International Conference on Learning Representations, 2022.
> >
> >
> > **Q3. Insufficient ablation study on \alpha. \alpha is only set to 1e-4, 1e-1, 5e-1 in section 5.4 with a large gap between 1e-4 and 1e-1. The author is recommended to provide more values of \alpha, at least 1e-2 and 1e-3. 4. Minor issues. The x label of Figure 2, Depth (T) rather than Time (T).**
> >
> > **Reply:** Thanks for your suggestion. In Figure 2, we have provided additional ablation study results for $\alpha$ values for both datasets and also fixed the labels of the figures.
> >
> > -----
> > We hope we have cleared your concerns about our work. We have also revised our manuscript according to your comments, and we would appreciate it if we can get your further feedback at your earliest convenience.

---

> > > ### Author Response · Authors · 2022-11-27
> > > **New Empirical Results on the OGBN-arXiv Node Classification Task. Any Questions from Reviewer 3KbW?**
> > >
> > > Dear reviewer,
> > >
> > > We would like to thank the reviewer again for your thoughtful reviews and valuable feedback.
> > >
> > > We have conducted additional experiments on the OGBN-arXiv node classification task [1] for different depths $T$ to further confirm the advantage of our DeepGRAND over the GRAND baseline [2] and GRAND++ [3]. The results in Table 1 below show that our DeepGRAND improves over the GRAND baseline and GRAND++ for all of the $T$ values. Furthermore, DeepGRAND’s standard deviations for the test accuracies in all cases are significantly lower than those of GRAND and GRAND++, indicating the more stable performance of our DeepGRAND on large benchmarks.
> > >
> > > Table 1: DeepGRAND, GRAND [2], and GRAND++ [3]'s test accuracies on the OGBN-arXiv benchmark [1] using multiple depths $T$.
> > >
> > > | Time | DeepGRAND-l      | GRAND-l          | GRAND++-l        |
> > > | ---- | ---------------- | ---------------- | ---------------- |
> > > | 1    | **69.78 $\pm$ 0.24** | 68.50 $\pm$ 0.76 | 68.79 $\pm$ 0.35 |
> > > | 4    | **70.45 $\pm$ 0.22** | 69.53 $\pm$ 0.21 | 69.68 $\pm$ 0.38 |
> > > | 6    | **70.51 $\pm$ 0.23** | 69.46 $\pm$ 0.43 | 69.71 $\pm$ 0.24 |
> > > | 8    | **70.41 $\pm$ 0.28** | 69.44 $\pm$ 0.30 | 69.61 $\pm$ 0.28 |
> > > | 32   | **69.44 $\pm$ 0.16** | 67.44 $\pm$ 0.59 | 69.41 $\pm$ 0.53 |
> > >
> > > **References**
> > >
> > > [1] “Node Property Prediction.” Open Graph Benchmark. Accessed November 27, 2022. https://ogb.stanford.edu/docs/nodeprop/#ogbn-arxiv.
> > >
> > > [2] Chamberlain, Benjamin Paul, James Rowbottom, Maria Gorinova, Stefan Webb, Emanuele Rossi, and Michael M. Bronstein. “Grand: Graph Neural Diffusion.” arXiv.org, September 22, 2021. https://arxiv.org/abs/2106.10934.
> > >
> > > [3]Thorpe, Matthew, Tan Minh Nguyen, Hedi Xia, Thomas Strohmer, Andrea Bertozzi, Stanley Osher, and Bao Wang. “Grand++: Graph Neural Diffusion with a Source Term.” OpenReview, September 29, 2021. https://openreview.net/forum?id=EMxu-dzvJk.
> > >
> > >
> > > -----
> > > We would appreciate it if you could let us know if there are additional questions or concerns about our revision and rebuttal. We would be happy to do any follow-up discussion or address any additional comments.

---

### Official Review · Reviewer_spFT · 2022-10-25

**Confidence:** 4
**Correctness:** 4
**Technical Novelty And Significance:** 2
**Empirical Novelty And Significance:** 2
**Recommendation:** 3

**Clarity, Quality, Novelty And Reproducibility:**

Clarity: good
Quality: good
Novelty : borderline
Reproducibility:good


**Strength And Weaknesses:**

### Strength
- This paper gives an excellent theoretical analysis
- The language is relatively standard
- The experimental results are relatively high

### Weaknesses
- The method is not original enough.
- They didn't fully explain the difference with Grand

**Summary Of The Paper:**

It is somehow incremental work based on GRAND. An objective of diffusion is quite simple, whether continuous or discrete. This paper states that it can alleviate over-smoothing, but there is not enough evidence to show the alleviating.

**Summary Of The Review:**

It is somehow incremental work based on GRAND.

---

> ### Author Response · Authors · 2022-11-18
> **Response to Reviewer spFT**
>
> Thank you for your thoughtful review and valuable feedback. Below we address your concerns.
>
> -----
> **Q1. The method is not original enough. They didn't fully explain the difference with Grand. It is somehow incremental work based on GRAND. An objective of diffusion is quite simple, whether continuous or discrete.**
>
> **Reply:** We believe there is a misunderstanding of the novelty of our DeepGRAND. Please allow us to clear this misunderstanding by clarifying the key technical contributions and advantages of DeepGRAND over GRAND. DeepGRAND takes advantage of the diffusion dynamic on graphs and modifies the diffusion equation in such a way that enables it to better learn deep and complex graph interactions. While GRAND closely followed the formulation of a pure diffusion process (equation (4)), DeepGRAND deviates from this by introducing the perturbing constant $\epsilon$ and the data-dependent scaling term $<X(t)>^\alpha$. With these modifications, the dynamic of DeepGRAND given by equation (9) no longer has the form of a pure diffusion process like GRAND, while still retaining its diffusive characteristics: at every infinitesimal instance, the information from graph nodes is aggregated for use in updating feature representation. This is important for two reasons: 1) diffusion and moderate smoothing behaviors have been shown to benefit GNN performance [1, 2]; 2) Pure diffusion causes over-smoothing (Section 3).
>
> The theoretical analysis in Section 4.2 (and Proposition 3 in particular) suggests DeepGRAND is more stable and resilient to over-smoothing thanks to these modifications. The intuition behind them is that $\epsilon$ improves stability by strengthening the feature boundedness property, while the scaling term slows down the convergence rate. As over-smoothing has been broadly described as the exponential convergence of node representations, the property that the dynamic induced by DeepGRAND is bounded by polynomial-like terms (equation (10)) illustrates the inherent resiliency that DeepGRAND possesses against the over-smoothing issue.
>
> **References**
>
> [1] Johannes Gasteiger, Stefan Weißenberger, and Stephan Gunnemann. Diffusion improves graph learning. Conference on Neural Information Processing Systems (NeurIPS), 2019.
>
> [2] Qimai Li, Zhichao Han, and Xiao-Ming Wu. Deeper insights into graph convolutional networks for semi-supervised learning, 2018. URL https://arxiv.org/abs/1801.07606
>
> -----
> **Q2. This paper states that it can alleviate over-smoothing, but there is not enough evidence to show the alleviating.**
>
> **Reply:** Thanks for your comment. We provide empirical results for the robustness of DeepGRAND using two ablation studies. First, we compared the test accuracies of DeepGRAND with other GNN methods and variants of GRAND when the depth $T$ increases in Table 2. The experimental results show that our method outperforms other methods in most cases across all datasets. In the second ablation study, we compare the performance of DeepGRAND with other methods when trained with different numbers of labeled nodes per class (Table 3). The experimental results show that on Cora, Citeseer, and Pubmed, DeepGRAND achieves better performance than most of the baseline methods while being on par with the others when the numbers of labeled nodes are 1, 2, 5, 10, and 20. However, for the later 3 benchmarks (CoauthorCS, Computers, Photo), DeepGRAND’s performance is worse than the baseline methods. This is because, in the ablation study with different label rates, we reused the depth $T$ tuned for the linear case of GRAND. Therefore, we have provided additional experimental results for this ablation study with $T$ values that were specifically fine-tuned for DeepGRAND in Table 4 in Appendix E. We also provided the specific list of $T$ values for each benchmark in Table 5. The new experimental results suggest that DeepGRAND significantly outperforms other methods across all benchmarks under different label rates. Furthermore, the standard deviations of the test accuracies are remarkably lower than those of the baseline methods, indicating that DeepGRAND is more stable than the baselines in a low label rate regime.
>
> -----
> We hope we have cleared your concerns about our work. We have also revised our manuscript according to your comments, and we would appreciate it if we can get your further feedback at your earliest convenience.

---

> > ### Author Response · Authors · 2022-11-27
> > **New Empirical Results on the OGBN-arXiv Node Classification Task. Any Questions from Reviewer spFT?**
> >
> > Dear reviewer,
> >
> > We would like to thank the reviewer again for your thoughtful reviews and valuable feedback.
> >
> > We have conducted additional experiments on the OGBN-arXiv node classification task [1] for different depths $T$ to further confirm the advantage of our DeepGRAND over the GRAND baseline [2] and GRAND++ [3]. The results in Table 1 below show that our DeepGRAND improves over the GRAND baseline and GRAND++ for all of the $T$ values. Furthermore, DeepGRAND’s standard deviations for the test accuracies in all cases are significantly lower than those of GRAND and GRAND++, indicating the more stable performance of our DeepGRAND on large benchmarks.
> >
> > Table 1: DeepGRAND, GRAND [2], and GRAND++ [3]'s test accuracies on the OGBN-arXiv benchmark [1] using multiple depths $T$.
> >
> > | Time | DeepGRAND-l      | GRAND-l          | GRAND++-l        |
> > | ---- | ---------------- | ---------------- | ---------------- |
> > | 1    | **69.78 $\pm$ 0.24** | 68.50 $\pm$ 0.76 | 68.79 $\pm$ 0.35 |
> > | 4    | **70.45 $\pm$ 0.22** | 69.53 $\pm$ 0.21 | 69.68 $\pm$ 0.38 |
> > | 6    | **70.51 $\pm$ 0.23** | 69.46 $\pm$ 0.43 | 69.71 $\pm$ 0.24 |
> > | 8    | **70.41 $\pm$ 0.28** | 69.44 $\pm$ 0.30 | 69.61 $\pm$ 0.28 |
> > | 32   | **69.44 $\pm$ 0.16** | 67.44 $\pm$ 0.59 | 69.41 $\pm$ 0.53 |
> >
> > **References**
> >
> > [1] “Node Property Prediction.” Open Graph Benchmark. Accessed November 27, 2022. https://ogb.stanford.edu/docs/nodeprop/#ogbn-arxiv.
> >
> > [2] Chamberlain, Benjamin Paul, James Rowbottom, Maria Gorinova, Stefan Webb, Emanuele Rossi, and Michael M. Bronstein. “Grand: Graph Neural Diffusion.” arXiv.org, September 22, 2021. https://arxiv.org/abs/2106.10934.
> >
> > [3]Thorpe, Matthew, Tan Minh Nguyen, Hedi Xia, Thomas Strohmer, Andrea Bertozzi, Stanley Osher, and Bao Wang. “Grand++: Graph Neural Diffusion with a Source Term.” OpenReview, September 29, 2021. https://openreview.net/forum?id=EMxu-dzvJk.
> >
> >
> > -----
> > We would appreciate it if you could let us know if there are additional questions or concerns about our revision and rebuttal. We would be happy to do any follow-up discussion or address any additional comments.

---

### Official Review · Reviewer_UGaV · 2022-11-03

**Confidence:** 2
**Clarity, Quality, Novelty And Reproducibility:** This paper is of fair novelty and qua…
**Correctness:** 2
**Technical Novelty And Significance:** 2
**Empirical Novelty And Significance:** 2
**Recommendation:** 5

**Strength And Weaknesses:**

In this paper, a continuous depth map neural network based on the diffusion process on the graph。

**Summary Of The Paper:**

In this paper, a continuous depth map neural network based on the diffusion process on the graph, DeepGRAND neural network, is proposed. It makes use of the scale term dependent on data and the disturbance to the diffusion rate of the graph, so that the real part of all the eigenvalues of the diffusion rate Marcus becomes negative, thereby alleviating the problem of over smoothing and ensuring the stability of the model. This paper empirically proves that DeepGRAND is superior to many existing graph neural networks in various graph depth learning benchmark tasks.

**Summary Of The Review:**

My research field is not related to this manuscript and I am not very familiar with the research content.

---

> ### Author Response · Authors · 2022-11-18
> **Response to Reviewer UGaV**
>
> Thanks for your review. The reviewer gave us an overall score of 5 (marginally below the acceptance threshold) with a score of 2 for correctness, technical novelty, and empirical novelty without giving us any particular reason. We would appreciate it if we can get more detailed feedback from the reviewer at your earliest convenience.

---

> > ### Author Response · Authors · 2022-11-27
> > **New Empirical Results on the OGBN-arXiv Node Classification Task. Any Questions from Reviewer UGaV?**
> >
> > Dear reviewer,
> >
> > We would like to thank the reviewer again for your thoughtful reviews and valuable feedback.
> >
> > We have conducted additional experiments on the OGBN-arXiv node classification task [1] for different depths $T$ to further confirm the advantage of our DeepGRAND over the GRAND baseline [2] and GRAND++ [3]. The results in Table 1 below show that our DeepGRAND improves over the GRAND baseline and GRAND++ for all of the $T$ values. Furthermore, DeepGRAND’s standard deviations for the test accuracies in all cases are significantly lower than those of GRAND and GRAND++, indicating the more stable performance of our DeepGRAND on large benchmarks.
> >
> > Table 1: DeepGRAND, GRAND [2], and GRAND++ [3]'s test accuracies on the OGBN-arXiv benchmark [1] using multiple depths $T$.
> >
> > | Time | DeepGRAND-l      | GRAND-l          | GRAND++-l        |
> > | ---- | ---------------- | ---------------- | ---------------- |
> > | 1    | **69.78 $\pm$ 0.24** | 68.50 $\pm$ 0.76 | 68.79 $\pm$ 0.35 |
> > | 4    | **70.45 $\pm$ 0.22** | 69.53 $\pm$ 0.21 | 69.68 $\pm$ 0.38 |
> > | 6    | **70.51 $\pm$ 0.23** | 69.46 $\pm$ 0.43 | 69.71 $\pm$ 0.24 |
> > | 8    | **70.41 $\pm$ 0.28** | 69.44 $\pm$ 0.30 | 69.61 $\pm$ 0.28 |
> > | 32   | **69.44 $\pm$ 0.16** | 67.44 $\pm$ 0.59 | 69.41 $\pm$ 0.53 |
> >
> > **References**
> >
> > [1] “Node Property Prediction.” Open Graph Benchmark. Accessed November 27, 2022. https://ogb.stanford.edu/docs/nodeprop/#ogbn-arxiv.
> >
> > [2] Chamberlain, Benjamin Paul, James Rowbottom, Maria Gorinova, Stefan Webb, Emanuele Rossi, and Michael M. Bronstein. “Grand: Graph Neural Diffusion.” arXiv.org, September 22, 2021. https://arxiv.org/abs/2106.10934.
> >
> > [3]Thorpe, Matthew, Tan Minh Nguyen, Hedi Xia, Thomas Strohmer, Andrea Bertozzi, Stanley Osher, and Bao Wang. “Grand++: Graph Neural Diffusion with a Source Term.” OpenReview, September 29, 2021. https://openreview.net/forum?id=EMxu-dzvJk.
> >
> >
> > -----
> > We would appreciate it if you could let us know if there are additional questions or concerns about our revision and rebuttal. We would be happy to do any follow-up discussion or address any additional comments.

---

### Author Response · Authors · 2022-11-18
**General Response (1)**

Dear AC and reviewers,

Thanks for your thoughtful reviews and valuable comments, which have helped us improve the paper significantly. We are encouraged by the endorsements that: 1) Our paper gives an excellent theoretical analysis (Reviewer spFT, 3KbW); 2) Fully designed experiments show the performance of the proposed method systematically(Reviewer spFT, 3KbW). We have updated our submission based on the reviewers' feedback, and we have highlighted our revision in blue.

Two main concerns from the reviewers are: (1) our method is not original enough and (2) the experimental results are not significantly improved compared with GRAND . We address these concerns here.

(1) **Novelty:** We believe there is a misunderstanding of the novelty of our DeepGRAND. Please allow us to clear this misunderstanding by clarifying the key technical contributions and advantages of DeepGRAND over GRAND. DeepGRAND takes advantage of the diffusion dynamic on graphs and modifies the diffusion equation in such a way that enables it to better learn deep and complex graph interactions. While GRAND closely followed the formulation of a pure diffusion process (equation (4)), DeepGRAND deviates from this by introducing the perturbing constant $\epsilon$ and the data-dependent scaling term $<X(t)>^\alpha$. With these modifications, the dynamic of DeepGRAND given by equation (9) no longer has the form of a pure diffusion process like GRAND, while still retaining its diffusive characteristics: at every infinitesimal instance, the information from graph nodes are aggregated for use in updating feature representation. This is important for two reasons: 1) diffusion and moderate smoothing behaviors have been shown to benefit GNN performance [1, 2]; 2) Pure diffusion causes over-smoothing (Section 3).

The theoretical analysis in Section 4.2 (and Proposition 3 in particular) suggests DeepGRAND is more stable and resilient to over-smoothing thanks to these modifications. The intuition behind them is that $\epsilon$ improves stability by strengthening the feature boundedness property, while the scaling term slows down the convergence rate. As over-smoothing has been broadly described as the exponential convergence of node representations, the property that the dynamic induced by DeepGRAND is bounded by polynomial-like terms (equation (10)) illustrates the inherent resiliency that DeepGRAND possesses against the over-smoothing issue.

---

> ### Author Response · Authors · 2022-11-18
> **General Response (2)**
>
> (2) **Experimental Results:** We believe there is a confusion between GRAND and GRAND++-l. Please allow us to clear this confusion by clarifying the difference between GRAND and GRAND++-l and the empirical advantages of our DeepGRAND over these two methods. GRANT++-l modifies GRAND by adding source terms to the original dynamics to keep the diffusion in GRAND from saturating [3]. Our DeepGRAND outperforms GRAND in all settings in Tables 1, 2, and 3 in the main text and in Table 4 in the Appendix. Our DeepGRAND and GRAND++-l are two orthogonal efforts that try to mitigate over-smoothing, i.e. feature saturation, in GRAND. Our DeepGRAND outperforms GRAND++-l  in most of the depth settings in Tables 1 and 2. However, we observe that GRAND++-l  becomes more stiff and unstable when the $T$ values are high, i.e., greater depth,  causing the ODE solver fails to converge. For example, GRAND++-l fails on Pubmed with $T=64$, $T=128$, on Citeseer with $T=128$, and on CoauthorCS with $T=32$.
>
> In Table 3 in our previous manuscript, the experimental results show that on Cora, Citeseer, and Pubmed, DeepGRAND achieves better performance than most of the baseline methods while being on par with the other when the number of labeled nodes are 1, 2, 5, 10, and 20. However, for the later 3 benchmarks (CoauthorCS, Computers, Photo), DeepGRAND’s performance is worse than the baseline methods. This is because, in this experiment, we reused the depth $T$ tuned for the linear case of GRAND. Therefore, we have provided additional experimental results for this ablation study with $T$ values that were specifically fine-tuned for DeepGRAND in Table 3 in the revised manuscript.  In Table 4 in Appendix E of our revision, we compare the results of GRAND-l, GRAND-nl, GRAND++-l, the old experiments of DeepGRAND, and the fine-tuned version of DeepGRAND. We provided the specific list of $T$ values for DeepGRAND for each benchmark in Table 5. The new experimental results suggest that DeepGRAND significantly outperforms other methods including GRAND++-l across all benchmarks under different label rates. Furthermore, the standard deviations of the test accuracies are remarkably lower than those of the baseline methods, indicating that DeepGRAND is more stable than the baselines in a low label rate regime.
>
> **References**
>
> [1] Johannes Gasteiger, Stefan Weißenberger, and Stephan Gunnemann. Diffusion improves graph learning. Conference on Neural Information Processing Systems (NeurIPS), 2019.
>
> [2] Qimai Li, Zhichao Han, and Xiao-Ming Wu. Deeper insights into graph convolutional networks for semi-supervised learning, 2018. URL https://arxiv.org/abs/1801.07606
>
> [3] Matthew Thorpe, Tan Minh Nguyen, Hedi Xia, Thomas Strohmer, Andrea Bertozzi, Stanley Osher, and Bao Wang. GRAND++: Graph neural diffusion with a source term. In International Conference on Learning Representations, 2022.
>
>
> -----
>
> We are glad to answer any further questions you have on our submission.

---

### Author Response · Authors · 2022-11-18
**Summary of Revision**

Incorporating the comments and suggestions from all reviewers, besides fixing typos and notations, we have made the following main changes in the revised paper.

1. We have revised section 4 to better explain our novel contribution compared to GRAND.

2. We have conducted hyper-parameters tuning for the optimal $T$ values specific for DeepGRAND and used those $T$ values to update the experiment results in the ablation study for multiple label rates. We compare the experiments of our method with optimal $T$ values and other variants of GRAND in appendix E, table 4, and a specific list of $T$ values that we used in table 5.

3. We have provided additional ablation study results for more $\alpha$ values and updated the visualization in figure 2.

---

### Author Response · Authors · 2022-11-18
**Any Questions from the Reviewers before the Deadline to Update Our Draft?**

Dear reviewers,

We would like to thank all reviewers again for your thoughtful reviews and valuable feedback. We have updated our manuscript and added new replies to your comments and questions with our latest experimental results. We have summarized the changes we made in the manuscript in the Summary of Revision below.

We would appreciate it if you could let us know if there are additional questions or concerns about our revision and rebuttal.

Best regards,

Authors

---

### Author Response · Authors · 2022-11-27
**Any Questions from the Reviewers? New Empirical Results on the OGBN-arXiv Node Classification Task**

Dear reviewers,

We would like to thank all reviewers again for your thoughtful reviews and valuable feedback.

We have conducted additional experiments on the OGBN-arXiv node classification task [1] for different depths $T$ to further confirm the advantage of our DeepGRAND over the GRAND baseline [2] and GRAND++ [3]. The results in Table 1 below show that our DeepGRAND improves over the GRAND baseline and GRAND++ for all of the $T$ values. Furthermore, DeepGRAND’s standard deviations for the test accuracies in all cases are significantly lower than those of GRAND and GRAND++, indicating the more stable performance of our DeepGRAND on large benchmarks.

Table 1: DeepGRAND, GRAND [2], and GRAND++ [3]'s test accuracies on the OGBN-arXiv benchmark [1] using multiple depths $T$.

| Time | DeepGRAND-l      | GRAND-l          | GRAND++-l        |
| ---- | ---------------- | ---------------- | ---------------- |
| 1    | **69.78 $\pm$ 0.24** | 68.50 $\pm$ 0.76 | 68.79 $\pm$ 0.35 |
| 4    | **70.45 $\pm$ 0.22** | 69.53 $\pm$ 0.21 | 69.68 $\pm$ 0.38 |
| 6    | **70.51 $\pm$ 0.23** | 69.46 $\pm$ 0.43 | 69.71 $\pm$ 0.24 |
| 8    | **70.41 $\pm$ 0.28** | 69.44 $\pm$ 0.30 | 69.61 $\pm$ 0.28 |
| 32   | **69.44 $\pm$ 0.16** | 67.44 $\pm$ 0.59 | 69.41 $\pm$ 0.53 |

**References**

[1] “Node Property Prediction.” Open Graph Benchmark. Accessed November 27, 2022. https://ogb.stanford.edu/docs/nodeprop/#ogbn-arxiv.

[2] Chamberlain, Benjamin Paul, James Rowbottom, Maria Gorinova, Stefan Webb, Emanuele Rossi, and Michael M. Bronstein. “Grand: Graph Neural Diffusion.” arXiv.org, September 22, 2021. https://arxiv.org/abs/2106.10934.

[3]Thorpe, Matthew, Tan Minh Nguyen, Hedi Xia, Thomas Strohmer, Andrea Bertozzi, Stanley Osher, and Bao Wang. “Grand++: Graph Neural Diffusion with a Source Term.” OpenReview, September 29, 2021. https://openreview.net/forum?id=EMxu-dzvJk.


-----
We would appreciate it if you could let us know if there are additional questions or concerns about our revision and rebuttal. We would be happy to do any follow-up discussion or address any additional comments.

---

### Decision · Program_Chairs · 2023-01-20

**Decision:**

Reject

**Justification For Why Not Higher Score:**

Insufficient novelty: the phenomenon and similar/other approaches to mitigate it were already done to some extent by the same authors of GRAND

**Justification For Why Not Lower Score:**

N/A

**Metareview: Summary, Strengths And Weaknesses:**

The paper derives a GNN architecture by discretizing a diffusion-type differential equation. This is a variant of a previously proposed graph neural diffusion (GRAND) paper of Chamberlain et al., which the authors show suffering from oversmoothing (as defined by Rusch et al.) This per se is not novel and has been noted by the authors themselves in subsequent works. At the same time, the advantage of having deep networks is not convincingly shown (rather, the performance of both GRAND and DeepGRAND degrade with depth, though the latter to a lesser extent). In the light of the above, we recommend rejection.

**Summary Of Ac-Reviewer Meeting:**

N/A